# RABGAP1 is a sensor that facilitates the sorting and processing of amyloid precursor protein

Jessica Eden[1], Jonathan G G Kaufman[1], Conceição Pereira [ID][1], Eleanor Fox [ID][1], Jerome Cattin-Ortolá[2], Lorena Benedetti [ID][3], Bart Nieuwenhuis [ID][1], David J Owen [ID][1], Jennifer Lippincott-Schwartz[3], Sean Munro [ID][2] & David C Gershlick [ID][1 ✉]

## Abstract

A hallmark of Alzheimer's disease (AD) is the accumulation of extracellular amyloid-β plaques in the brain. Amyloid-β is a 40–42 amino acid peptide generated by proteolytic processing of amyloid precursor protein (APP) via membrane-bound proteases. APP is a transmembrane protein, and its trafficking to sites of proteolysis represents a rate-limiting step in AD progression. Although APP processing has been well-studied, its trafficking itinerary and machinery remain incompletely understood. To address this, we performed an unbiased interaction screen for interactors of the APP cytosolic tail. We identified previously characterised APP binders as well as novel interactors, including RABGAP1. We demonstrated that RABGAP1 partially co-localises with APP and directly interacts with a YENPTY motif in the APP cytosolic tail. Depletion or overexpression of RABGAP1 caused mistrafficking and misprocessing of endogenous APP in human and rodent neurons. This effect is dependent on the GAP activity of RABGAP1, demonstrating that RABGAP1 affects the trafficking of APP by modulating RAB activity on endosomal subdomains. This novel trafficking mechanism has implications for other NPXY cargoes and presents a possible therapeutic avenue to explore.

**Keywords** APP; Alzheimer's Disease; Trafficking; RABGAP1
**Subject Categories** Membranes & Trafficking; Molecular Biology of Disease; Organelles

## Introduction

Amyloid precursor protein (APP) is a type-I membrane-spanning protein, which plays a central role in Alzheimer's disease (AD) pathology. APP is synthesised in the endoplasmic reticulum (ER), before being transported to the Golgi apparatus for further post-translational modification, including O- and N-linked glycosylation (Wang et al, 2017). From the Golgi apparatus, APP can traffic directly to the plasma membrane, or it can enter the endolysosomal

system. During its complex trafficking itinerary, APP encounters several secretase enzymes that are capable of cleaving APP and producing a unique set of APP peptides. One of these peptides is the toxic amyloid-β fragment that accumulates extracellularly in the brains of AD patients.

There are two well-characterised canonical processing pathways of APP: the non-amyloidogenic and the amyloidogenic pathway. In the healthy brain, most APP processing takes place through the non-amyloidogenic pathway, which avoids the production of the toxic amyloid-β peptide (Sisodia et al, 1990; Agostinho et al, 2015). Here, the first cleavage event is mediated by an α-secretase, such as the 'A Disintegrin and Metalloproteinase' (ADAM) family of enzymes, that cleaves APP within the amyloid-β sequence. This generates a large ectodomain fragment, called sAPPα, as well as a smaller membrane-bound carboxyl-terminal fragment, C83 (Haass et al, 2012; Wang et al, 2017). Following α-secretase cleavage, the C83 fragment undergoes an intramembrane cut by the sequential action of the γ-secretase complex. This process yields a 3 kDa soluble peptide, p3, on the lumenal side, and the APP intracellular domain (AICD) peptide on the cytoplasmic side (Haass et al, 1993). In the amyloidogenic pathway, the first cleavage event is mediated by a β-secretase enzyme, BACE1. BACE1 can cleave the lumenal domain of APP 99 amino acids away from the C-terminus, between residues Met596 and Asp597 of the APP695 isoform (residue Asp1 of the amyloid-β peptide sequence). This releases a sAPPβ fragment for secretion, whilst the carboxyl-terminal fragment, C99, remains embedded in the membrane (Haass et al, 2012). After BACE1 cleavage, sequential cleavage by the γ-secretase complex releases amyloid-β on the lumenal/extracellular side and the AICD fragment on the cytoplasmic side. The α-secretase enzymes are generally believed to reside at the plasma membrane, although some studies have suggested that a pool of α-secretase may also exist at the *trans*-Golgi network (TGN) (Sisodia, 1992; Parvathy et al, 1999; Sambamurti et al, 1992; Skovronsky et al, 2000). At steady state, active BACE1 is predominantly found in the endosomes, where a lower pH (around pH 4.5) is optimal for β-secretase activity (Vassar et al, 1999), although some studies have also reported BACE1 activity at the TGN (Thinakaran et al, 1996; Xu et al, 1997). The active γ-secretase complex appears to be widely distributed throughout the secretory pathway, with several studies showing that it localises to the ER and Golgi (Zhang et al, 1998), as

[1]Cambridge Institute for Medical Research, University of Cambridge, Cambridge, UK. [2]MRC laboratory of Molecular Biology, Francis Crick Avenue, Cambridge, UK. [3]Janelia Research Campus, Howard Hughes Medical Institute, Ashburn, VA, USA. ✉E-mail: dg553@cam.ac.uk

well as the endolysosomal system and plasma membrane (Chyung et al, 2005; Walter et al, 1996; Maltese et al, 2001; Kanatsu et al, 2014).

The post-Golgi trafficking itinerary of APP has complex molecular determinants that are still largely unresolved. The highest concentrations of APP at steady state are found primarily at the TGN and early endosomes (Caporaso et al, 1994; Choy et al, 2012; Toh et al, 2017). Some studies have demonstrated that APP processing can occur at the Golgi, where α- and β-secretases compete with each other for APP processing (Xu et al, 1997; Choy et al, 2012; Toh et al, 2017). Only 10% of APP is localised to the plasma membrane at any given time, with over 50% re-internalised within 10 min (Koo et al, 1996). APP is rapidly endocytosed from the plasma membrane due to the presence of a carboxyl YENPTY motif in its cytosolic tail.

After budding from the plasma membrane, clathrin-coated vesicles (CCVs) fuse with the early endosomes to deliver APP. A fraction of APP can then be recycled back to the plasma membrane via recycling endosomes. Most of it, however, remains in the endolysosomal system where it is cleaved, before undergoing either lysosomal or proteasomal-mediated degradation (Tam et al, 2014; El Ayadi et al, 2012). A subpopulation of APP is believed to undergo direct Golgi-to-endosome trafficking, mediated by an interaction between the cytosolic tail of APP with different CCV adaptors, including AP-1, AP-4 and GGAs (Perez et al, 1999; Burgos et al, 2010). The endosomes appear to serve as the primary site for amyloid-β production.

The trafficking of APP is directed by a variety of molecular machinery. This includes cytosolic adaptors such as munc18-interacting (MINT) proteins, FE65 and NUMB (Caster and Kahn, 2013; Shrivastava-Ranjan et al, 2008; Kyriazis et al, 2008; Borg et al, 1996), as well as the sortilin-related receptor, SORLA, a type-I membrane protein that mediates the retrograde transport of APP, potentially in cooperation with the retromer complex (Fjorback et al, 2012). While the comprehensive molecular machinery that governs APP distribution through the secretory pathway and endolysosomal system remains only partially understood, it plays a vital role in determining APP localisation within different subcellular compartments. This localisation influences the co-localisation of APP with various processing enzymes, directly affecting the production of amyloid-β peptide.

In this study, we used an unbiased mass spectrometry approach to identify novel interactors of the APP tail. Here, we identified RABGAP1, a protein known to play an important role in membrane remodelling but not previously reported to interact with APP. We demonstrate that the phosphotyrosine binding (PTB) domain of RABGAP1 can directly bind to the APP cytosolic tail through its YENPTY motif. This interaction is essential to mediate cleavage of endogenous APP through the amyloidogenic pathway in both human iPSC-derived cortical glutamatergic neurons (i3 neurons) and primary rat hippocampal neurons. Upon CRISPRi knockdown (KD) of RABGAP1 in i3 neurons, levels of C99, the precursor to the toxic amyloid-β peptide, are significantly reduced. C99 abundance is restored to physiological levels upon complementation with WT RABGAP1, but not with mutants of RABGAP1 that either lack GAP activity or can no longer bind to APP. Inversely, when RABGAP1 is overexpressed in i3 neurons, an accumulation of C99 is observed. This study provides clear evidence that RABGAP1 abundance mediates the amyloidogenic

processing of APP in the endosomes. This mechanism likely applies to the trafficking of other endosomal NPXY-containing cargoes.

# Results

## Identification of novel APP interactors

The cytosolic tail of APP is only 47 amino acids long; however, it is rich in trafficking motifs that can bind to a variety of proteins. The proteins that interact with the APP tail play a critical role in determining its subcellular localisation and processing fate. As demonstrated in previous studies, mutations to the APP tail itself, or to proteins that interact with the tail, can perturb the processing of APP (Januário et al, 2022; Burgos et al, 2010). To identify novel interactors of the APP tail, an unbiased interaction screen was carried out. The WT APP tail was fused to Glutathione S-transferase (GST) and affinity chromatography was carried out using HEK293T cell lysates (Fig. 1A). A variety of proteins were enriched in the affinity chromatography carried out with GST-fused APP tail, compared to GST only (Fig. 1B–D; Dataset EV1). This included many known APP interactors, such as the retriever complex component, sorting nexin 17 (SNX17) (Lee et al, 2008); MINT1/2 (also called APBA1/2) and NUMB, cytosolic adaptors known to mediate APP endocytosis (Sullivan et al, 2014; Roncarati et al, 2002; Kyriazis et al, 2008) and FE65 (also called APBB1), another cytosolic adaptor that mediates APP trafficking and has been suggested to form a transcriptionally active complex with the AICD (Cao and Südhof, 2001). Analysis by a Panther GO over-representation test revealed that proteins implicated in retromer sorting were enriched by over 100-fold (Fig. 1C). Similarly, proteins associated with tubular endosomes were enriched by 49-fold. This suggested that the mass spectrometry data contained proteins involved in the endosomal sorting of APP. Interestingly, several novel putative APP interactors were also identified in the mass spectrometry data, including RABGAP1, NCKAP1 and PDLIM7.

To identify specific residues within the APP tail that are important for binding the interactors identified in Fig. 1, affinity chromatography was repeated, as described above, using a range of APP tail mutants (Fig. 2A). Here, point mutations were introduced within key consensus motifs within the APP tail (previously described in Januário et al, 2022) and their binding was validated by immunoblotting. Figure 2B shows that the binding of several key interactors of the WT APP tail was successfully validated using this method. None of the protein hits bound GST alone, verifying the specificity of their interaction with the APP tail. For RABGAP1, binding to the APP tail was abolished in mutants Y682A, N684A and Y687A (Fig. 2C). This corresponds to the YENPTY motif in the far C-terminus of the APP tail. SNX17 binding was also reduced in the Y682A and Y687A mutants and completely abolished in the N684A and FFE689-9AAA mutants (Fig. 2D). Interestingly, whilst the FFE689-91 residues appeared to be essential for SNX17 binding, the FFE689-91AAA mutant bound RABGAP1 with approximately a 3-fold increase. We hypothesise that this is due to a liberation in the binding of a competitive factor (perhaps SNX17) in the FFE689-91AAA mutant compared to the WT tail. For NUMB, binding was abolished in D664A, Y682A, N684A and Y687A mutants (Fig. 2E). PDLIM7 was able to bind all APP tail mutants,

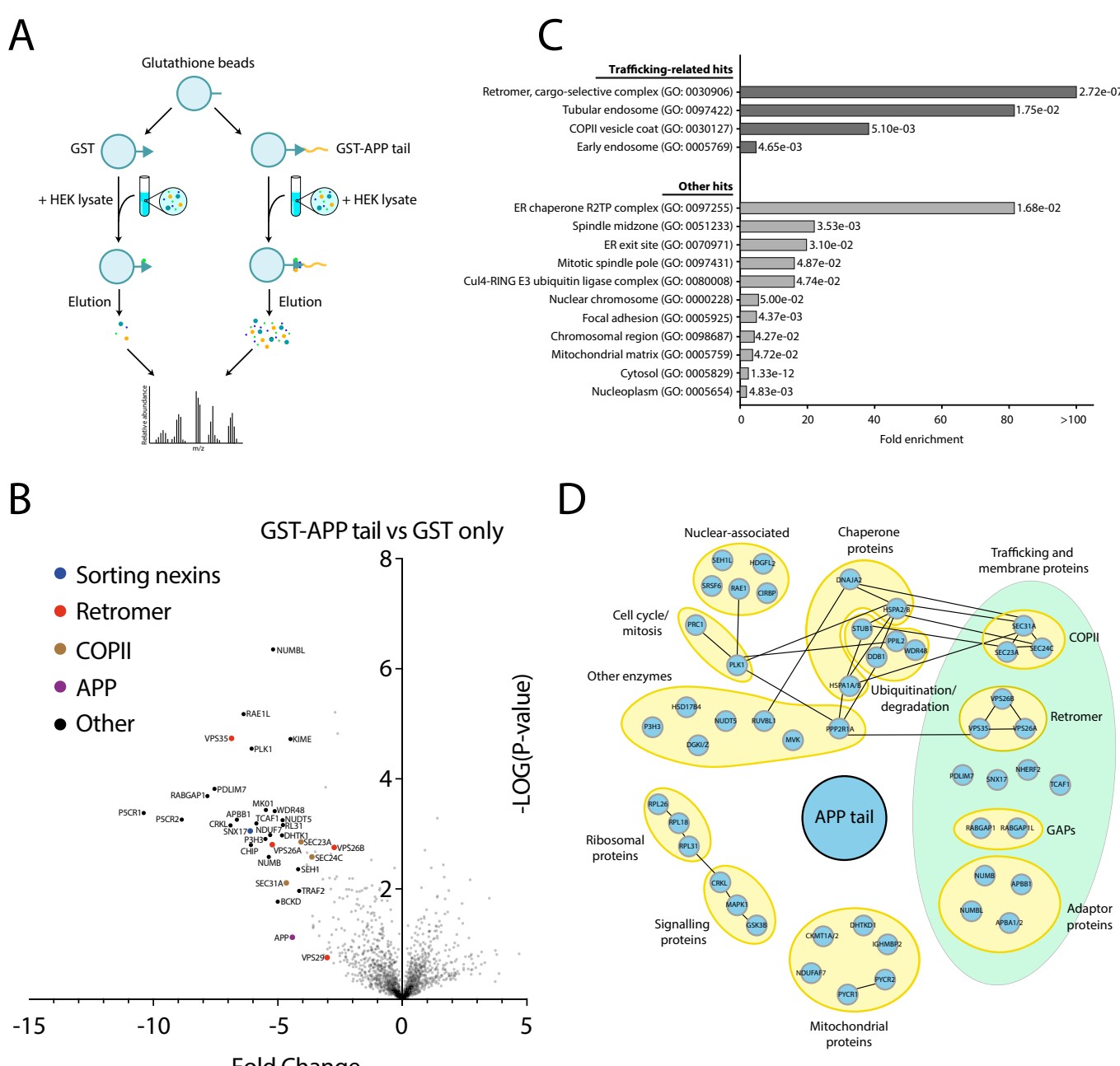

**Figure 1. Identification of novel interactors of the APP cytosolic tail using affinity chromatography.**

(A) A schematic showing the workflow of the affinity chromatography experiment, using either GST fused to the cytosolic tail of APP, or GST only. (B) A volcano plot showing the mass spectrometry analysis of proteins interacting with the cytosolic tail of APP. Affinity chromatography was carried out using HEK293T lysate in combination with bacterially-expressed GST fused to the APP tail, or GST alone. The mean spectral intensities of bound proteins from three independent experiments are plotted. $N = 3$ biological repeats. Full mass spectrometry dataset included in Dataset EV1. (C) Panther GO analysis of mass spectrometry hits from affinity chromatography using GST-APP tail. The fold enrichment is displayed for each GO cellular component category of a PANTHER overrepresentation test (Fisher's exact), along with the corresponding corrected P value (false discovery rate, FDR). In all, 101 hits were included in the analysis, where the enrichment was ≥ 2-fold. GO categories with a FDR P value < 0.05 were included. (D) Interaction map of the APP tail with proteins from HEK293T lysate, identified by mass spectrometry and grouped according to their cellular functions. All displayed proteins interacted with the APP tail. Positive interactions between different protein hits have been mapped as connected nodes. Note, for some protein isoforms, such as APBA1 and APBA2 (labelled APBA1/2), peptide contribution could not be distinguished from one another in the mass spectrometry and so both isoforms have been included in the label. Hits with $P ≤ 0.01$ were included for interaction mapping. Statistical analysis was performed using the Perseus module of MaxQuant (Tyanova et al, 2016). Source data are available online for this figure.

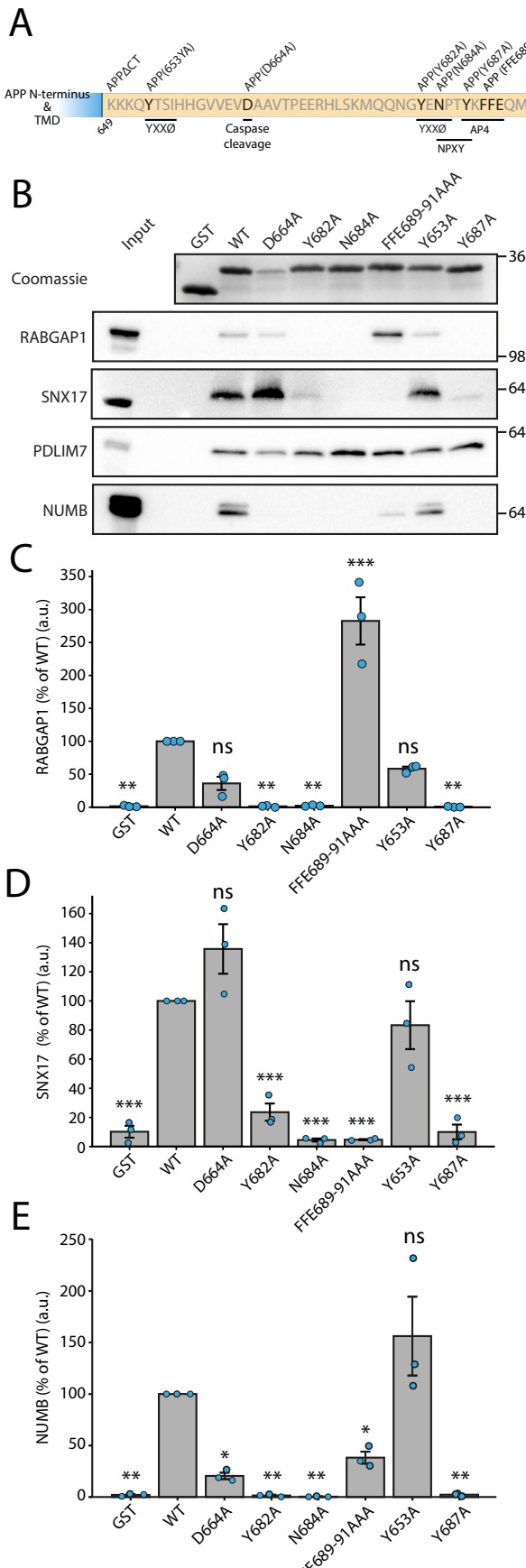

**Figure 2. Characterisation of the binding of novel interactors with the APP tail.**

(A) A schematic of point mutations introduced in the cytosolic tail of APP (649–695). Residues are numbered according to APP isoform 695. Important consensus motifs and binding sites have been highlighted. (B) Affinity chromatography from Fig. 1 was repeated using either GST fused to the APP WT tail, a series of APP tail mutants, or GST only. Eluates were used for immunoblotting and probed using antibodies against hits previously identified by mass spectrometry. (C) Quantification of RABGAP1, as shown in (B). The protein detected in each condition is expressed as a percentage of the WT tail. (D) Quantification of SNX17, as shown in (B). (E) Quantification of NUMB, as shown in (B). $N = 3$ biological repeats. Bars represent the mean ± standard error of the mean (SEM) (C–E). Statistical significance (C–E) was determined using one-way ANOVA followed by Tukey's Honest Significant Difference (HSD) multiple comparisons post-hoc test (FWER = 0.05). ***$P \leq 0.001$; **$P \leq 0.01$; *$P \leq 0.05$; ns not significant. Source data are available online for this figure.

suggesting that PDLIM7 binding to the APP tail may occur through a novel binding motif in the tail.

## The PTB domain of RABGAP1 directly binds to the YENPTY motif in the cytosolic tail of APP

RABGAP1, also known as GAPCenA or TBC1D11, is a cytosolic small GTPase-activating protein (GAP) consisting of 1069 amino acids, with a molecular mass of approximately 120 kDa. It was initially identified as the GAP for RAB6A at the Golgi apparatus (Cuif et al, 1999), although other studies have since suggested that it can act on other RABs, including RAB11, RAB4 and RAB36 (Fuchs et al, 2007; Kanno et al, 2010). RABGAP1 comprises three distinct domains: an N-terminal phosphotyrosine-binding (PTB) domain, a central catalytic TRE2/BUB2/CDC16 (TBC) domain, and a C-terminal coiled-coil domain. A previous study has suggested that RABGAP1 interacts with TUFT1 to promote perinuclear lysosomal clustering and enhanced mTORC1 signalling (Kawasaki et al, 2018). Samarelli et al (2020) demonstrated that, whilst RABGAP1 is mostly cytosolic at steady state, it also co-localises with RAB11A-positive recycling endosomes and RAB5-positive early endosomes, but does not co-localise with lysosomal markers RAB7 and LAMP1. This study also showed that the PTB domain of RABGAP1 can directly interact with the NPXY motif in the cytosolic tail of active β1-integrins, enabling the recycling of endocytosed receptors back to the plasma membrane through the attenuation of RAB11A (Samarelli et al, 2020). Because of this, it seemed plausible that the PTB domain of RABGAP1 may also interact with the YENPTY motif in the APP tail. To verify this, full length RABGAP1 was modelled with APP in AlphaFold3 (Fig. 3A–C) (Abramson et al, 2024). Indeed, the AlphaFold3 model predicted that the RABGAP1 PTB domain interacts with the YENPTY motif in the APP tail (NPXY pLDDT= 94.6).

To validate the AlphaFold3 model of APP binding to the PTB domain of RABGAP1, isothermal titration calorimetry (ITC) was performed using the purified PTB domain of RABGAP1 with two APP peptides. The first was the WT APP tail, starting from the MQQ sequence at position 677 (Fig. 2A). The second was a mutant form of the APP tail, where the YEN residues of the YENPTY motif were mutated to AAA (herein referred to as APPmut). Based on the AlphaFold3 model, it was predicted that the APPmut peptide would be unable to interact with the RABGAP1 PTB domain. ITC confirmed that the WT APP tail bound the PTB domain of

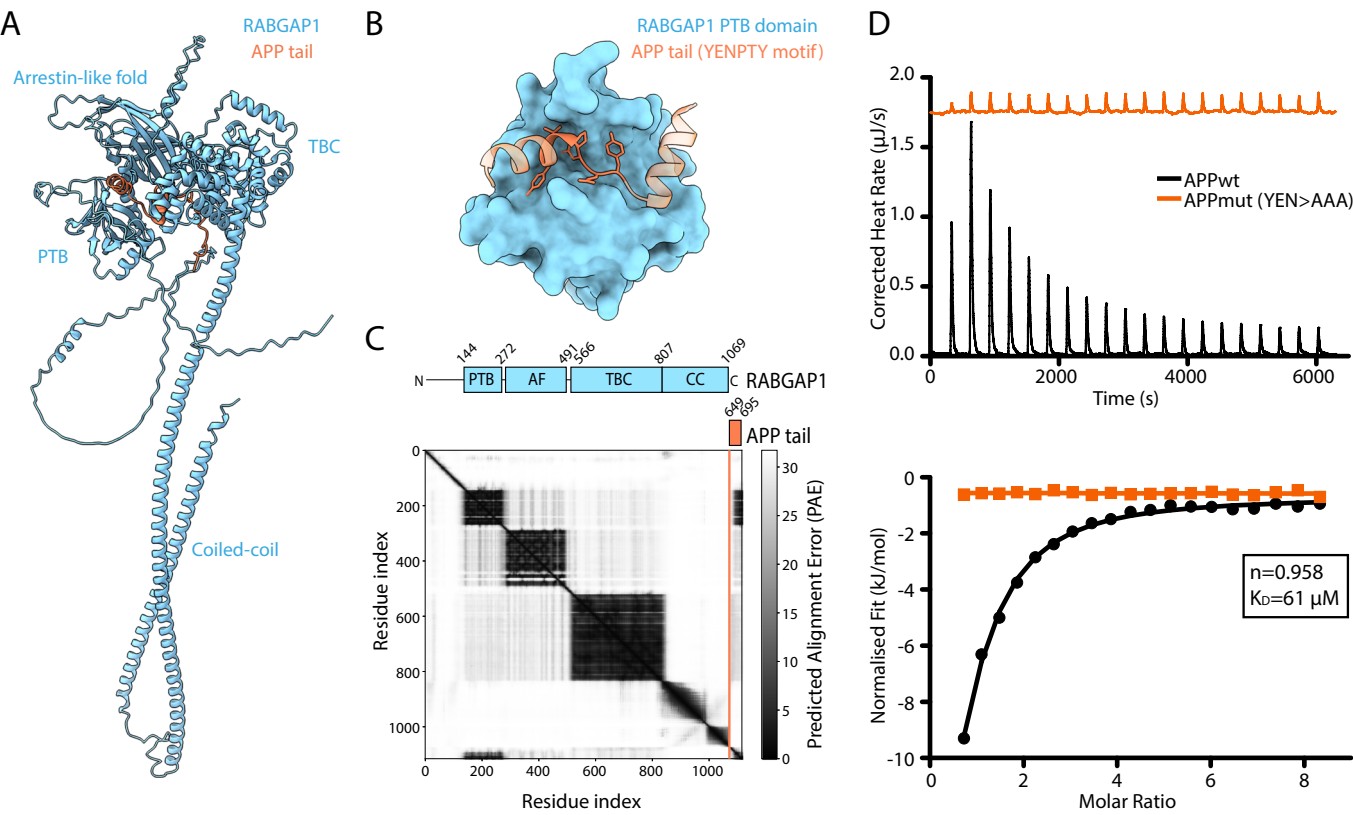

**Figure 3. The PTB domain of RABGAP1 is predicted to bind to the YENPTY motif of the APP tail.**

(A) AlphaFold3 modelling of full length RABGAP1 (blue) displayed bound to APP tail (orange). This hydrophobic interaction occurs via β-augmentation of the YEN residues (of the YENPTY motif). (B) The same model as (A), but showing only the PTB domain of RABGAP1. Atoms are shown for the YENPTY motif in the APP tail. (C) Predicted Alignment Error (PAE) plot of the AlphaFold3 model shown in (A, B). The domain architecture of RABGAP1 is depicted above its corresponding domain on the PAE plot. The domains are labelled as follows: PTB phosphotyrosine binding domain, AF arrestin-like fold, TBC Tre-2/Bub2/Cdc16 domain, CC coiled-coil. (D) Isothermal titration calorimetry (ITC) quantification of the binding between the PTB domain of RABGAP1 with either the WT APP tail (APPwt) or a mutant form where the YEN residues at positions 682–684 are mutated to AAA (APPmut). Source data are available online for this figure.

RABGAP1 with a $K_D$ of approximately 61 μM and a predicted stoichiometry of 1:1 (Fig. 3D). This binding was abolished upon mutation of the YEN residues in the APPmut peptide, demonstrating that the YENPTY motif is essential for the binding of the APP tail to the PTB domain of RABGAP1.

## RABGAP1 KO affects the processing of exogenously expressed APP in HeLa cells

To assess the role of RABGAP1 in the processing of APP, a transient RABGAP1 CRISPR knockout (KO) was generated in Cas9 HeLa cells stably expressing HaloTag-APP-mNeonGreen (Fig. 4A,B). Here, we used a dual-tagged APP reporter to assess the levels of both C-terminal fragments (CTFs) and N-terminal fragments (NTFs) of APP in living cells. The use of this dual-tagged APP reporter has been extensively characterised in Januário et al, 2022, demonstrating that the steady state distribution of APP is not detectably affected by the presence of the tags and that dual-tagged APP undergoes physiological processing by the secretase enzymes. Using this dual-tagged APP HeLa cell line, we have previously demonstrated that APP cleavage leads to secretion of the N-terminus to the extracellular space, whilst the CTFs are degraded intracellularly (Januário et al, 2022). Upon KO of RABGAP1,

flow cytometry analysis revealed a significant increase in the levels of both mNeonGreen and HaloTag fluorescence (Fig. 4C,D). This indicates that there is an increase in either full length APP or both CTFs and NTFs, suggesting that RABGAP1 KO reduces the processing of APP. There was also a small but significant increase in the ratio of mNeonGreen:HaloTag in the RABGAP1 KO cells (Fig. 4E). The KO of RABGAP1 was validated via immunoblotting (Fig. 4B) and it was also confirmed via qPCR that the levels of APP transcription were not upregulated upon RABGAP1 KO (Fig. 4F). To assess whether RABGAP1 KO induces general defects in the post-Golgi secretory pathway, we performed a kinetic trafficking assay based on the retention using selective hooks (RUSH) system, as described in Pereira et al (2023). This assay tracked the cell surface delivery of a model secretory cargo, LAMP1ΔYQTI, which traffics from the ER to the Golgi and ultimately to the plasma membrane. Following synchronised ER release, we monitored surface arrival in WT cells, single KOs of RABGAP1 or its homologue RABGAP1L, and a double KO of both genes (Appendix Fig. S10). RABGAP1L KO cells showed no impairment in post-Golgi trafficking, while RABGAP1 KO cells exhibited only a modest delay, suggesting that RABGAP1 plays a limited, non-essential role in global secretory trafficking.

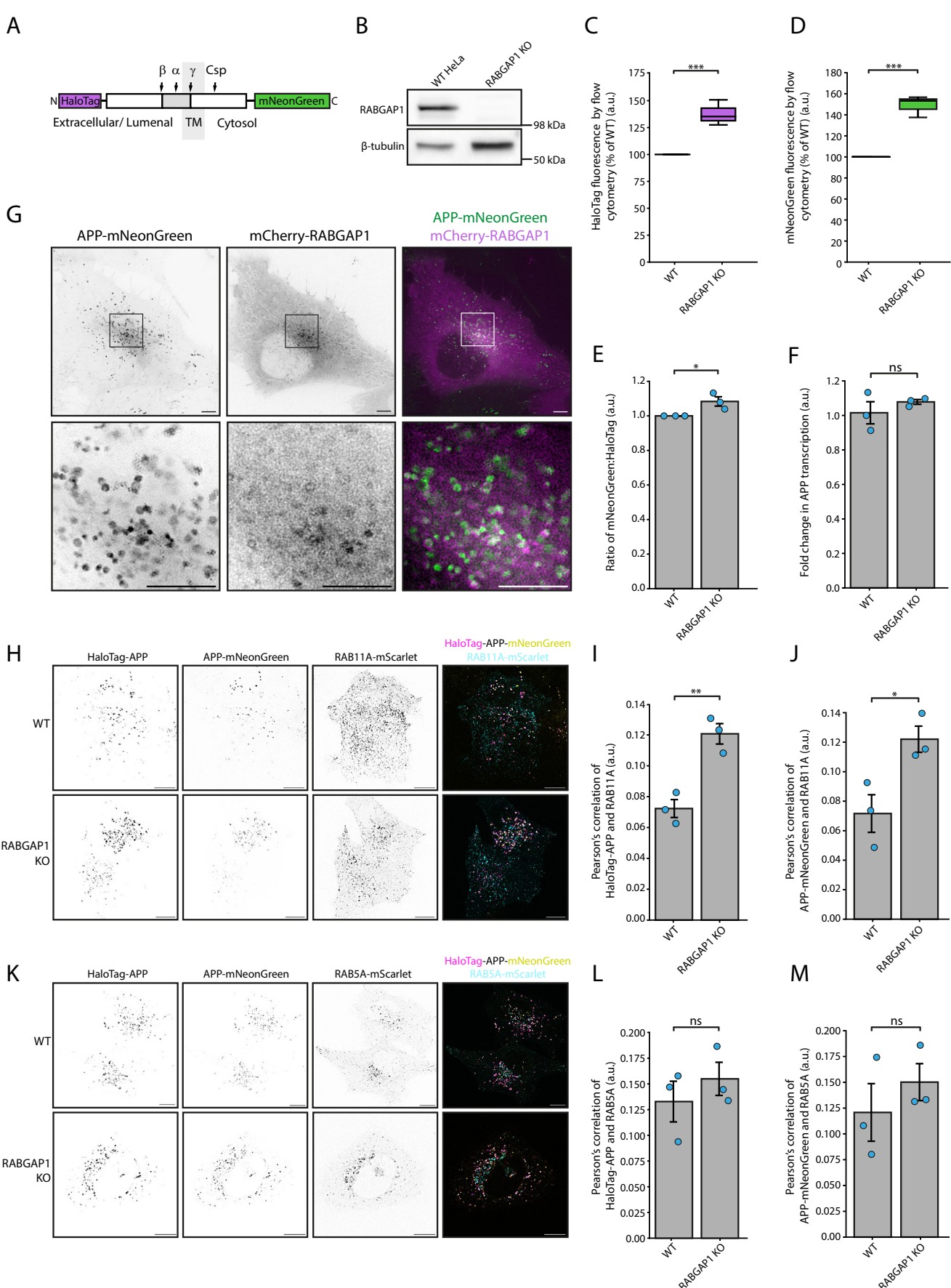

**Figure 4.  APP processing is perturbed upon RABGAP1 KO in HeLa cells.**

(A) Schematic of the dual-tagged APP construct, showing a HaloTag fused to the N-terminus of APP and an mNeonGreen tag fused to the cytosolic side. Major APP cleavage sites have been highlighted. (B) Immunoblotting validation of transient CRISPR/Cas9 knockout (KO) of RABGAP1 in HeLa cells. β-tubulin has been used as a loading control. (C) N-terminal HaloTag-JF646 fluorescence levels of WT HeLa and RABGAP1 KO cells, measured by flow cytometry. Fluorescence intensities are expressed as a percentage of WT HeLa cells. $N = 3$ biological repeats. (D) C-terminal mNeonGreen fluorescence levels of WT HeLa and RABGAP1 KO cells, measured by flow cytometry. $N = 3$ biological repeats. Box plots (C, D) display the median (centre line), the interquartile range (box bounds = 25th and 75th percentiles), and whiskers extending to the smallest and largest data points. (E) Ratio of mNeonGreen:HaloTag-JF646 fluorescence as quantified by flow cytometry. $N = 3$ biological repeats. (F) Quantitative real-time PCR (qPCR) of APP transcription in HeLa cells upon KO of RABGAP1. $N = 3$ biological repeats. (G) Co-localisation of APP and RABGAP1 in HeLa cells pre-treated with 25 µM DAPT for 24 h. Live cell structured illumination microscopy of steady state HaloTag-APP-mNeonGreen and mCherry-RABGAP1. Images were taken from Movies EV1 and 2. Images were taken on a Zeiss Elyra 7 Lattice SIM microscope. $N = 3$ biological repeats. Scale bars, 5 µm. (H) Live cell structured illumination microscopy of steady state HaloTag-APP-mNeonGreen and RAB11A-mScarlet in HeLa cells. Scale bars, 10 µm. (I) Quantification of imaging shown in (H). Pearson's correlation of HaloTag-APP and RAB11A-mScarlet. $N = 3$ biological repeats. Each point represents the average from one biological replicate. A minimum of 28 cells were analysed in total in each condition. (J) Quantification of imaging shown in (H). Pearson's correlation of APP-mNeonGreen and RAB11A-mScarlet. (K) Live cell structured illumination microscopy of steady state HaloTag-APP-mNeonGreen and RAB5A-mScarlet. Scale bars, 10 µm. (L) Quantification of imaging shown in (K). Pearson's correlation of HaloTag-APP and RAB5A-mScarlet. $N = 3$ biological repeats. Each point represents the average from one biological replicate. A minimum of 30 cells were analysed in total in each condition. (M) Quantification of imaging shown in (K). Pearson's correlation of APP-mNeonGreen and RAB5A-mScarlet. $N = 3$ biological repeats. Bars represent the mean ± SEM (E–M). Statistical significance (C–M) was assessed using an unpaired t test. ***$P \leq 0.001$; **$P \leq 0.01$; *$P \leq 0.05$; ns not significant. Source data are available online for this figure.

To determine if APP and RABGAP1 co-localise to the same subcellular compartments, mCherry-RABGAP1 was transiently overexpressed in HeLa cells stably overexpressing HaloTag-APP-mNeonGreen (Fig. 4G; Movies EV1 and 2). In line with previous data, RABGAP1 was largely cytosolic, but could also be visualised on a subpopulation of APP-positive puncta. APP showed a partial co-localisation with both RAB5A and RAB11A, consistent with its steady state localisation in both early and recycling endosomes (Fig. 4H–M). In RABGAP1 KO cells, a significant increase in co-localisation was observed between N-terminal HaloTag and RAB11A-mScarlet (Fig. 4I), as well as between C-terminal mNeonGreen and RAB11A-mScarlet (Fig. 4J). This suggested that, upon RABGAP1 KO, both N- and C-terminal APP fragments, or full length APP, accumulate in a RAB11A-positive compartment. There was no significant change in the co-localisation of either N-terminal or C-terminal APP fragments with RAB5A (Fig. 4K–M).

In summary, RABGAP1 binds to the cytosolic tail of APP in vitro and, in HeLa cells, KO of RABGAP1 causes a significant increase in the co-localisation of APP with RAB11A, suggesting that RABGAP1 is important for the endosomal sorting and subsequent cleavage of APP.

## RABGAP1 KD affects the processing of endogenous APP in a neuronal model

The experiments conducted so far in HeLa cells showed that RABGAP1 may play a role in APP processing. Next, we wanted to assess this role using a more physiologically relevant cell type, such as neurons, where the endogenous expression levels of both APP and its processing enzymes are significantly different to HeLa cells. In particular, the expression levels of BACE1 are very low in HeLa cells and, consequently, most processing takes place through the non-amyloidogenic processing pathway.

To assess APP processing in a more physiologically relevant model, a two-pronged approach was employed, using both primary rat hippocampal neurons and i3 neurons. We first confirmed that, like HeLa cells, APP primarily localised to a subpopulation of endosomes in i3 neurons. Here, overexpressed HaloTag-APP-mNeonGreen and mScarlet-FYVE, a marker for early endosomes, was used to enable live cell high-resolution imaging (Fig. 5A,B;

Movie EV3). CRISPRi was then used to generate RABGAP1 KD i3 neurons and the KD efficiency was assessed via immunoblotting (Fig. 5E,F). qPCR was also used to confirm that the transcription of APP was unaffected in the RABGAP1 KD neurons compared to WT (Fig. 5C). WT and RABGAP1 KD i3 neurons were differentiated and on day 15, the neurons were lysed and used for immunoblotting to assess the processing of endogenous APP. Here, we selected the Cell Signalling Technologies D54D2 antibody (CST-XP, 8243) as it recognises an epitope at the far N-terminus of the amyloid-β peptide, enabling detection of C99 and the amyloid-β peptide, but not C83 (Fig. 5D). Alongside this, a C-terminal APP antibody (Invitrogen, 51-2700) was also used to detect full length endogenous APP (Fig. 5D). This antibody binds to an epitope in the far C-terminus of APP, within the AICD fragment.

Immunoblotting using the D54D2 antibody revealed clear differences in the processing of endogenous APP between the WT and RABGAP1 KD neurons (Fig. 5E–I; Appendix Fig. S1). In the RABGAP1 KD neurons, there was a significant decrease in the abundance of a 16 kDa peptide, corresponding to the approximate size of the C83/C99 processing fragment (Fig. 5E,H). As this band was detected using the D54D2 antibody, which binds C99 but not C83, it was determined that this 16 kDa band was likely to be C99. This band was also detected using another C99-specific antibody, 82E1, where it was also less abundant in the RABGAP1 KD neurons (Appendix Fig. S2). To further verify that this peptide was indeed C99, differentiated day 14 i3 neurons were treated with DAPT, a γ-secretase inhibitor, for 24 h before lysis on day 15 (Appendix Fig. S3). DAPT prevents subsequent processing of C83 and C99 by γ-secretase, trapping this fragment in the membrane after initial α- or β-secretase cleavage. As expected, an accumulation of the 16 kDa fragment was detected in DAPT-treated neurons, compared to untreated neurons, confirming that this band was indeed C99. Notably, upon γ-secretase inhibition, there was still a significant reduction in the 16 kDa fragment abundance in the RABGAP1 KD neurons compared to WT (Appendix Fig. S3). This demonstrated that the processing defect in RABGAP1 KD neurons occurs upstream of γ-secretase cleavage, suggesting that RABGAP1 is important for mediating initial APP cleavage by either α- or β-secretase. Moreover, this effect was not observed upon CRISPRi KD of the RABGAP1 homologue, RABGAP1L, demonstrating the specificity for RABGAP1 in this process (Appendix Fig. S3). In

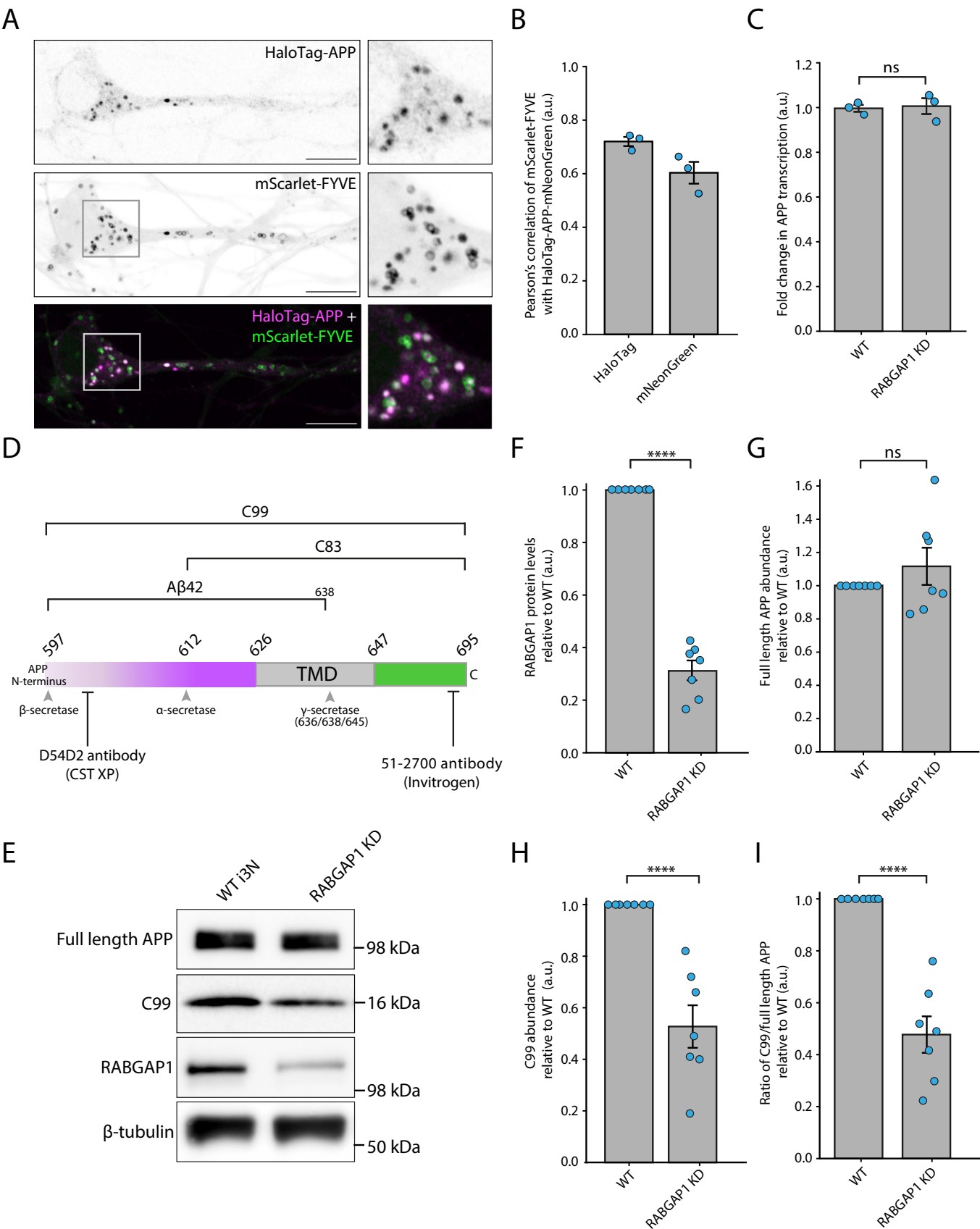

**Figure 5. Endogenous C99 production is significantly decreased in RABGAP1 KD i3 neurons.**

(A) Co-localisation of APP and endosomes in day 15 i3 neurons. Live cell imaging of steady state HaloTag-APP-mNeonGreen and mScarlet-FYVE. Images were taken from Movie EV3. Images were taken on a Zeiss LSM880 Airyscan microscope. Scale bars, 10 μm. (B) Quantification of imaging shown in (A). Pearson's correlation of mScarlet-FYVE and both HaloTag and mNeonGreen from HaloTag-APP-mNeonGreen expression. $N = 3$ biological repeats. Each point represents the average from one biological replicate. A minimum of 30 cells were analysed in total in each condition. (C) qPCR of APP transcription in WT and RABGAP1 KD day 15 i3 neurons. $N = 3$ biological repeats. (D) A schematic of the C-terminus of APP, showing secretase cleavage sites and antibody epitopes. The β-amyloid CT695 (Invitrogen, 51-2700) binds to the far C-terminus of APP and detects both C99 and C83. The D54D2 antibody (Cell Signalling Technologies, 8243) binds to the N-terminus of the amyloid-β peptide and can therefore only detect C99 and not C83. (E) Immunoblotting of endogenous APP processing in WT and RABGAP1 KD i3 neurons. Neurons were lysed at day 15 of differentiation. APP processing levels were assessed using the D54D2 antibody (Cell Signalling Technologies, 8243) which detects C99, but not C83, as well as the β-amyloid CT695 (Invitrogen, 51-2700) antibody to detect full length APP levels. β-tubulin was used as a loading control. RABGAP1 was probed to monitor RABGAP1 KD efficiency. (F) Quantification of RABGAP1 KD efficiency, as seen in (E). Protein abundance is normalised to WT levels. $N = 7$ biological repeats. (G) Quantification of full length APP (110 kDa), as seen in (E). Full length APP was probed using the β-amyloid CT695 (Invitrogen, 51-2700) antibody. $N = 7$ biological repeats. (H) Quantification of C99 abundance (16 kDa), as seen in (E). C99 was probed using the D54D2 antibody (Cell Signalling Technologies, 8243) antibody. $N = 7$ biological repeats. (I) Quantification of the ratio of C99 (16 kDa) over full length APP (110 kDa), as shown in (E). Bars represent the mean ± SEM (B–I). Statistical significance (C–I) was assessed using an unpaired t test. $N = 7$ biological repeats. ****$P \leq 0.0001$; ns = not significant. Source data are available online for this figure.

addition to this, there was also a small but significant increase in an unidentified processing fragment at ~64 kDa, as well as a significant decrease in a peptide at ~100 kDa, which is likely to be the sAPPα fragment (Appendix Fig. S2).

The significant reduction in the 16 kDa peptide fragment was also detected using the APP C-terminal antibody (Invitrogen, 51-2700) that recognises an epitope in the cytosolic AICD fragment (Appendix Figs. S4 and S8). In agreement with the data generated using the CST-XP D54D2 antibody, a significant decrease in a 16 kDa CTF was observed in RABGAP1 KD neurons compared to WT neurons. This decrease was still observed when the neurons were pre-treated with either DAPT, a γ-secretase inhibitor, or MG132, a proteasomal inhibitor, confirming that the RABGAP1 KD does not increase proteasomal degradation of APP CTFs (Appendix Figs. S4 and S8).

In parallel to the studies conducted in i3 neurons, the role of RABGAP1 in endogenous APP processing was also assessed in embryonic day 18 (E18) hippocampal rat neurons. Consistent with the data generated using the i3 neuronal model, shRNA-mediated abrogation of RABGAP1 in primary hippocampal neurons resulted in a significant decrease in the abundance of CTFs (Appendix Fig. S5). Here, two different CTF peptides were detected using the APP C-terminal antibody (Invitrogen, 51-2700), and both decreased in abundance, although these were both of a larger molecular weight compared to the 16 kDa peptide detected in i3 neurons (Appendix Figs. S5 and S9A). No signal was detected using the CST-XP D54D2 antibody in primary rat hippocampal neurons, perhaps due to species differences in the APP protein sequence. In summary, two independent neuronal models have been used to demonstrate that RABGAP1 plays an essential role in mediating the processing of APP.

## Exogenously expressed WT RABGAP1 restores levels of C99, but a GAP-deficient RABGAP1 mutant does not

To demonstrate the specificity of the phenotype observed upon RABGAP1 KD, a complementation experiment was carried out to test whether re-introduction of RABGAP1 could recover the loss of C99 observed in RABGAP1 KD i3 neurons (Fig. 6A,B; Appendix Fig. S7A). Here, the recovery was tested using WT RABGAP1, along with several RABGAP1 mutants. This included a RABGAP1 GAP-deficient mutant, where a R612A point mutation introduced into the IxxDxxR arginine finger motif of RABGAP1 abolishes its

GAP activity (Samarelli et al, 2020). Additionally, based on AlphaFold3 modelling of the interaction between RABGAP1 PTB domain and the APP tail, a triple point mutation (L216S, S236F, A264F) was also used to abolish binding to the APP tail (Fig. 6E). To test whether the expression of WT RABGAP1, or the RABGAP1 mutants, could recover the loss of C99, new stable iPSC lines were generated and subsequently differentiated. The complementation of WT RABGAP1 into the RABGAP1 KD neurons successfully restored C99 abundance to a level comparable with WT neurons. This was not achieved when using the RABGAP1 GAP-deficient mutant (R612A) or the RABGAP1 mutant (L216S, S236F, A264F) that is predicted to be unable to bind to the APP tail. In both conditions, C99 levels remained the same as those observed upon RABGAP1 KD, or upon complementation with an empty vector (negative control). This provides direct evidence that RABGAP1 can bind to the APP tail in cells and disrupt its processing through the amyloidogenic pathway. It also clearly demonstrates that the role of RABGAP1 in APP trafficking is dependent on its GAP activity, suggesting that it mediates this effect through RAB protein inactivation.

## Overexpression of WT RABGAP1 increases levels of C99 in i3 neurons

APP processing was also assessed upon overexpression of either WT RABGAP1 or the mutants described above (Fig. 6C,D; Appendix Fig. S7B). Here, stable overexpression iPSC lines were generated, differentiated to day 15 and subsequently lysed for immunoblotting. Overexpression of WT RABGAP1 caused a significant increase in C99 levels. This was not observed upon overexpression of either the RABGAP1 GAP-deficient mutant (R612A) or the RABGAP1 binding mutant (L216S, S236F, A264F), where the level of C99 remained comparable to WT neurons and cells transfected with the empty overexpression vector. This data supports a model where the expression levels of RABGAP1 directly correlate to the level of processing through the amyloidogenic pathway.

Overexpression of RABGAP1 was also assessed in primary rat hippocampal neurons (Appendix Figs. S6 and S9B). Here, over-expression of RABGAP1 by approximately 10-fold was achieved using lentivirus transduction of RABGAP1-mEmerald. This caused a significant shift in the ratio between the two CTFs compared to WT processing. Here, a significant and consistent decrease in the

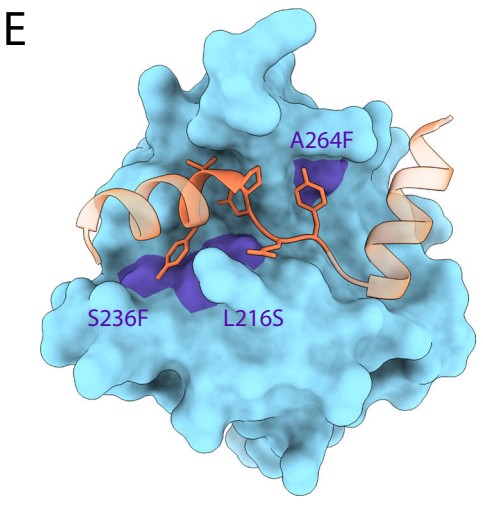

**A** RABGAP1 KD + recovery condition

WT i3N, RABGAP1 KD, + WT RABGAP1, + Empty Vector, + RABGAP1 R612A, + RABGAP1 L216S, S236F, A264F

APP C99 — 16 kDa
RABGAP1 — 98 kDa
β-tubulin — 50 kDa

**B**

ns
ns
ns
**
**
**
**

C99 abundance (relative to recovery with WT RABGAP1) (a.u.)

RABGAP1 KD only
+ WT RABGAP1
+ Empty Vector
+ RABGAP1 R612A
+ RABGAP1 L216S, S236F, A264F

RABGAP1 KD + recovery condition

**C** RABGAP1 overexpression

WT i3N, + WT RABGAP1, + Empty Vector, + RABGAP1 R612A, + RABGAP1 L216S, S236F, A264F

APP C99 — 16 kDa
RABGAP1 — 98 kDa
β-tubulin — 50 kDa

**D**

ns
ns
ns
**
**
**
**

C99 abundance (relative to WT RABGAP1 overexpression) (a.u.)

WT
+ WT RABGAP1
+ Empty Vector
+ RABGAP1 R612A
+ RABGAP1 L216S, S236F, A264F

RABGAP1 overexpression

**E**

A264F
S236F    L216S

**Figure 6.   Complementation with WT RABGAP1 can recover the loss of C99 in RABGAP1 KD i3 neurons.**

(A) Stable iPSC lines were generated where the RABGAP1 KD line was complemented with either WT RABGAP1, RABGAP1 GAP-deficient mutant (R612A) or a RABGAP1 mutant that is predicted to no longer interact with the APP tail (L216S, S236F, A264F). A negative control line was also made using an empty vector. The cell lines were differentiated, lysed on day 15 and subsequently used for immunoblot analysis of APP processing. The D54D2 antibody (Cell Signalling, 8243) was used to assess levels of CTF C99. RABGAP1 levels were also probed for, along with β-tubulin that was used as a loading control. $N = 4$ biological repeats. (B) Quantification of C99 levels shown in (A). Levels of C99 are expressed relative to the WT RABGAP1 recovery condition. (C) Stable iPSC lines were generated with overexpression of either WT RABGAP1, a RABGAP1 GAP-deficient mutant (R612A) or a RABGAP1 mutant that is predicted to no longer interact with the APP tail (L216S, S236F, A264F). The cell lines were differentiated and C99 levels were assessed, as described in (A). $N = 3$ biological repeats. (D) Quantification of C99 levels shown in (C). Bars represent the mean ± SEM (B–D). Statistical significance (B–D) was determined using one-way ANOVA followed by Tukey's Honest Significant Difference (HSD) multiple comparisons post-hoc test (FWER = 0.05). **$P \le 0.01$; ns = not significant. (E) Predicted interaction interface between RABGAP1 PTB domain and the YENPTY motif of the APP tail, modelled in AlphaFold3. The binding of the PTB domain of RABGAP1 (blue) to the APP tail (orange) is predicted to be abolished upon triple mutation of A264F, L216S and S236F (purple). Source data are available online for this figure.

abundance of the lower molecular weight CTF (annotated as smaller CTF) was observed (Appendix Fig. S6C). This was coupled with an increase in the levels of the higher molecular weight CTF, although this increase was more variable and not statistically significant (Appendix Fig. S6D). This indicates that, like RABGAP1 KD, its overexpression also changes the processing of APP in primary rat neurons, providing further evidence for the direct role of RABGAP1 in mediating endogenous APP trafficking and cleavage.

## Discussion

An unbiased mass spectrometry screen identified several novel APP interactors, including PDLIM7 and RABGAP1. We have shown, using several in vitro techniques, that RABGAP1 can directly bind to the YENPTY motif in the APP tail, where residues YEN are essential for this binding. Using ITC, we demonstrated that the wild-type APP tail binds to RABGAP1's PTB domain with a $K_D$ of ~61 μM and a stoichiometry of 1:1 (Fig. 3D). This interaction affinity is consistent with the binding affinities observed for GAP-RAB interactions, which often fall within the micromolar range, as well as for various adaptor proteins that bind APP. For example, the μ4 subunit of AP-4 binds to the APP tail with a $K_D$ of ~$27.7 \pm 2.5$ μM (Ross et al, 2014). These relatively weak affinity interactions are functionally important, as they allow for the dynamic and reversible trafficking of APP.

The findings presented in this study demonstrate that RABGAP1 plays a direct role in mediating the trafficking and cleavage of APP in both iPSC-derived human i3 neurons and primary rat hippocampal neurons. Dysregulation of RABGAP1 levels, through either KD or overexpression, markedly disrupts the processing of APP to CTFs. In i3 neurons, RABGAP1 KD led to a significant reduction of the pathogenic amyloid-β precursor, C99. This is indicative of diminished amyloidogenic processing of APP. Notably, C99 levels were rescued by complementation with WT RABGAP1, but not by a GAP-deficient mutant of RABGAP1, demonstrating the importance of its GAP activity for the regulation of APP processing. In RABGAP1 KD i3 neurons, the reduction in C99 levels suggests a likely decrease in the pool of APP localised within early endosomes, where BACE1 activity is enriched. This suggests that in the absence of RABGAP1, APP and NPXY cargoes are misdirected to other subcellular localisations, away from BACE1.

In addition to RABGAP1, in this study we identified several other known and novel interactors of the APP cytosolic tail. Representative isoforms for two of the three members of the

retromer core complex (VPS35, VPS26B and VPS26A) were enriched in our proteomics. Retromer forms a complex with a variety of sorting nexins to mediate cargo selection and membrane tubulation for the sorting of cargo from endosomes (Bonifacino and Hurley, 2008; Seaman, 2005; van Weering et al, 2010; Lucas et al, 2016). APP has been implicated as a retromer cargo previously (Fjorback et al, 2012); however, it was unclear if this interaction was direct or if APP is associated with retromer via the transmembrane adaptor SORLA (Lane et al, 2010; Sullivan et al, 2011). Evidence generated in this study would support a direct interaction of APP with retromer.

Although no retromer-associated sorting nexins were identified in our mass spectrometry, we did identify SNX17. SNX17 has been linked to the retriever complex, a trimeric complex analogous to retromer, and is known to be important for endosome to plasma membrane recycling (McNally et al, 2017; Gershlick and Lucas, 2017). Interestingly, SNX17 interacts with NPXY motifs, and indeed, we demonstrate that SNX17 interacts with the NPXY in APP (Ghai et al, 2013). Moreover, the retriever complex has also been shown to interact with integrins via a SNX17-NPXY interaction (Steinberg et al, 2012; McNally et al, 2017). This mechanism could thus be a conserved trafficking regulatory mechanism.

In addition to driving the trafficking of APP, the soluble AICD, cleaved from the membrane by γ-secretase, has been implicated in several downstream signalling events. Previous data has suggested that the AICD may translocate to the nucleus to regulate gene transcription. Interestingly, a subset of our interactors, identified by mass spectrometry, are indeed nuclear proteins, including HDGFL2, CIRBP, SEH1L and FE65. Other studies have shown that the AICD can interact with FE65 (also known as APBB1) via its NPXY motif, and this may generate a transcriptionally active complex with the histone acetyltransferase TIP60 (Cao and Südhof, 2001). Moreover, a protective function for the cold-inducible RNA-binding protein (CIRBP) against neuronal amyloid toxicity has also been described (Su et al, 2020).

To assess the role of RABGAP1 in the trafficking and cleavage of APP, preliminary studies were first carried out using dual-tagged overexpressed APP in HeLa cells (Fig. 4). Here, an accumulation of both N- and C-terminal tags was observed in RABGAP1 KO cells via flow cytometry, indicative of an increase in full length APP and a reduction in its processing. In addition to this, both APP tags showed a significant increase in their co-localisation with RAB11A upon RABGAP1 KO. These findings support a model in which the GAP activity of RABGAP1 is important for inactivating RAB11A and maintaining the localisation of APP within early endosomes,

where its cleavage takes place. In the absence of RABGAP1, increased recruitment of RAB11A to the endosomal membrane may lead to APP accumulation in a recycling endosomal compartment that is less permissive to its efficient processing. Future studies will be needed to further interrogate this mechanism and determine the precise molecular pathways by which RABGAP1 regulates APP trafficking and cleavage.

When endogenous APP processing was assessed in RABGAP1 KD i3 neurons, a reduction in β-secretase cleavage and C99 abundance was observed (Fig. 5). This observation generally agrees with the data generated in HeLa cells, as BACE1 is predominantly active within acidified early endosomes, and not within recycling endosomes. Interestingly, an accumulation of full length APP was not observed in i3 neurons, which could be because the expression profiles of the secretases are vastly different between HeLa cells and i3 neurons, as well as the fact that dual-tagged APP was overexpressed in HeLa cells, whereas in the i3 neurons endogenous APP processing was assessed. In primary rat hippocampal neurons, changes to endogenous APP cleavage and the abundance of CTFs were once again observed upon RABGAP1 KD (Appendix Fig. S5); however, here the CTFs detected were of a slightly larger molecular weight and we were unable to use the D54D2 antibody (Cell Signalling Technologies, 8243) to determine which fragment was the C99 peptide. Overall, whilst there were some differences in the phenotype observed in different cell models, as you may expect due to differences in the expression of APP and its secretases, RABGAP1 KD/KO and overexpression consistently resulted in aberrant APP processing.

We recognise that, while our findings provide important insights into APP trafficking, several limitations of the study should be considered. Firstly, to effectively inhibit γ-secretase cleavage and assess the accumulation of C99, we used a relatively high concentration of DAPT, which may not fully reflect physiological conditions. Secondly, while our data support a role for RABGAP1 in APP trafficking and processing, we cannot exclude potential non-specific or broader effects of RABGAP1 KO, particularly given its known involvement in general membrane trafficking. While we observed a clear effect of RABGAP1 on RAB11A activity and localisation, we cannot rule out the possibility that RABGAP1 also acts as a GAP for other RAB proteins, which may contribute to the phenotypes observed. Alternatively, our RABGAP1 cargo binding mutation (L216S, S236F, A264F) may affect the binding of RABGAP1 to multiple cargoes. We also note that our initial experiments were conducted in HeLa cells expressing dual-tagged, overexpressed APP. To address this, we validated our key findings in two independent neuronal models, where endogenous APP trafficking was assessed without the use of exogenous tags. Finally, further studies will be needed to elucidate the precise molecular mechanisms by which RABGAP1 regulates APP sorting and to determine whether its function extends to other NPXY-containing cargoes.

As RABGAP1 binds to the cytosolic tail of APP to mediate its trafficking and processing, one interpretation of the data would be that RABGAP1 is a trafficking adaptor. However, this is inconsistent with the importance of the GAP activity in maintaining the proper processing of APP. An alternative model would be that RABGAP1 acts instead as a sensor for NPXY cargo load within endosomes. RABGAP1 could then regulate the sorting of NPXY cargoes into distinct endosomal subdomains by remodelling the RAB profile on the membrane. This is likely to include the inactivation of RAB11A to prevent the maturation of an early to recycling endosome. This facilitates efficient sorting of NPXY cargoes to either the plasma membrane, to the lysosomes for degradation and for retention within the early endosomal system. This mechanism would enable RABGAP1 to finely tune the abundance of NPXY cargoes within early endosomes. There are 88 proteins in the revised and annotated human proteome that have a cytosolic-facing NPXY motif. Given that the PTB domain of RABGAP1 can directly bind to NPXY motifs, this could represent a generalised membrane sorting mechanism.

Given our data that shows abrogation or inhibition of RABGAP1 results in less amyloidogenic processing of APP, targeting RABGAP1 could be a promising strategy for therapeutics. However, biallelic loss of RABGAP1 in both humans and mouse models causes severe neurodevelopmental defects, including intellectual disabilities, global developmental delays, delayed myelination and loss of white matter volume in the brain (da Silva-Buttkus et al, 2023; Oh et al, 2022). In addition, therapeutics will need to consider that RABGAP1 has been demonstrated to be important for the trafficking of β-integrins via an NPXY interaction and is likely to be necessary for the efficient trafficking of other cargoes. Therefore, any future therapeutics targeting RABGAP1 will require, at a minimum, specific tissue targeting and well-controlled delivery to limit off-target effects.

# Methods

**Reagents and tools table**

| Reagent/resource | Reference or source | Identifier or catalog number |
|---|---|---|
| **Experimental models** | | |
| iNeuron (*H. sapiens*) | Ward lab (National Institutes of Health, USA) | |
| HeLa cells (*H. sapiens*) | ATCC | CCL2 |
| HEK293T (*H. sapiens*) | ATCC | CRL-3216 |
| T7 express *E. coli* | NEB | C2566H |
| **Recombinant DNA** | | |
| HaloTag-APP-mNeonGreen | Gershlick lab (Januário et al, 2022. PMID: 35753347) | |
| mCherry-RABGAP1 | Böttcher lab (Samarelli et al, 2020. PMID: 32843574) | |
| PiggyBac RABGAP1 complementation plasmids (WT, R612A, [L216S, S236F, A264F]) | This study | "Methods" |
| **Antibodies** | | |
| APP/C99 (D54D2) | Cell Signalling Technology | 8243 |
| APP/AICD | Invitrogen | 51-2700 |
| APP/C99 (82E1) | Stratech Scientific, 10323 | 10323 |
| RABGAP1 | Abcam | ab153992 |
| β-tubulin | ProteinTech | 66240-1-Ig |
| Alexa Fluor-conjugated secondary antibody | Invitrogen | A32728 |
| **Oligonucleotides and other sequence-based reagents** | | |
| HaloTag-fw primer | This study. Generated by IDT | "Methods" |
| HaloTag-rev primer | This study. Generated by IDT | "Methods" |
| RABGAP1-fw | This study. Generated by IDT | "Methods" |

| Reagent/resource | Reference or source | Identifier or catalog number |
|---|---|---|
| RABGAP1-rev | This study. Generated by IDT | "Methods" |
| RABGAP1 CRISPR guide 1 | This study. Generated by IDT | "Methods" |
| RABGAP1 CRISPR guide 2 | This study. Generated by IDT | "Methods" |
| RABGAP1 CRISPRi guide 1 | This study. Generated by IDT | "Methods" |
| RABGAP1 CRISPRi guide 2 | This study. Generated by IDT | "Methods" |
| RABGAP1L CRISPRi guide 1 | This study. Generated by IDT | "Methods" |
| RABGAP1L CRISPRi guide 2 | This study. Generated by IDT | "Methods" |
| APP-fw qPCR primer | This study. Generated by IDT | "Methods" |
| APP-rev qPCR primer | This study. Generated by IDT | "Methods" |
| TBP-fw qPCR primer | This study. Generated by IDT | "Methods" |
| TBP-rev qPCR primer | This study. Generated by IDT | "Methods" |
| RABGAP1 shRNA | Origene | TL706390 |
| **Chemicals, enzymes and other reagents** | | |
| JF646 HaloTag ligand | Levis lab (Janelia Research Campus, USA) | |
| SpeI-HF | NEB | R3133S |
| MluI-HF | NEB | R3198S |
| NheI-HF | NEB | R3131S |
| Q5 Hot Start High-Fidelity 2X Master Mix | NEB | M0494S |
| KLD Enzyme Mix | NEB | M0554S |
| DNase | Sigma | D5025-15KU |
| Fugene6 | Promega | E2691 |
| Lipofectamine 3000 | Invitrogen | L3000001 |
| Lipofectamine LTX | Invitrogen | A12621 |
| Lipofectamine STEM | Invitrogen | STEM00001 |
| DAPT | Abcam | 208255-80-5 |
| MG132 | Cell Signaling Technology | 2194 |
| Clarity ECL | Bio-Rad | 1705061 |
| WesternBright Sirius ECL | Advansta | K-12043-D10 |
| **Software** | | |
| FlowJo | Becton Dickinson and Company [BD] 2006–2020 | Version 10.7.1, for Mac OS X |
| AlphaFold3 | https://alphafoldserver.com/welcome Abramson J et al, 2024 | |
| ChimeraX | UCSF | Version 1.8 |
| Image Lab | Bio-Rad | Version 6.1 |
| ImageJ | https://imagej.net/ij/index.html | |
| **Other** | | |
| DMEM | Sigma | D6429 |
| MycoZap | Lonza | VZA-2012 |
| FBS | Merck | F7524 |
| Cell-Strainer-capped 5 ml round-bottom tubes | Corning | 352235 |
| Opti-MEM | Gibco | 31985047 |
| Lenti-X Concentrator | Takara Bio | 631232 |

| Reagent/resource | Reference or source | Identifier or catalog number |
|---|---|---|
| hESC-qualified matrigel | Corning | 354277 |
| Essential-8 medium | Gibco | A1517001 |
| Rho-associated protein kinase inhibitor Y-27632 (ROCKI) | Tocris Bioscience | 1254 |
| Accutase | Gibco | A1110501 |
| DMEM/F12 | Gibco | 11330032 |
| N2 100x | Gibco | 17502048 |
| Non-essential amino acids (NEAA) 100x | Gibco | 11140050 |
| GlutaMAX 100x | Gibco | 25030081 |
| Poly-L-Ornithine | Sigma | P3655 |
| BrainPhys Media | Stemcell Technologies | 05790 |
| B27 Supplement 50x | Gibco | 17504044 |
| BDNF | PeproTech | 450-02 |
| NT-3 | PeproTech | 450-03 |
| Laminin | Gibco | 23017015 |
| D-glucose | Sigma-Aldrich | G7021 |
| HBSS | Merck | H2387 |
| Papain dissociation powder | Lorne Labs | LK003178 |
| NeuroCult SM1 supplement 50x | Stemcell Technologies | 05711 |
| NeuroCult Neuronal Plating Medium | Stemcell Technologies | 05713 |

## Molecular biology

PiggyBac RABGAP1 complementation plasmids, used to test the recovery of the RABGAP1 KD phenotype in i3 neurons, contained either WT RABGAP1, a RABGAP1 GAP-deficient (R612A) mutant, RABGAP1 (L216S, S236F, A264F) mutant, or an empty piggyBac vector (negative control). The RABGAP1 point mutations were first introduced into a smaller CMV-mScarlet-RABGAP1 vector via PCR using mutation-specific primers. A KLD reaction was then carried out, as described previously. For the triple RABGAP1 (L216S, S236F, A264F) mutant, three successive rounds of PCR and KLD were carried out to introduce the 3 point mutations into the CMV-mScarlet-RABGAP1 vector.

A piggyBac backbone was designed to express the RABGAP1 mutants in the iPSCs for the complementation experiment. This transposon vector contained two separate expression cassettes: (1) RABGAP1 expressed under the CAG promoter, (2) A HaloTag and hygromycin resistance gene, separated by a T2A, under the expression of an EF1a promoter. The vector was designed this way to enable the positive selection of cells (through both hygromycin selection and FACS) without using a RABGAP1 fluorescent fusion protein.

The HaloTag and T2A were inserted into a piggyBac transposon vector, just before a hygromycin gene, via Gibson assembly. Here, PCR primers were designed to amplify the HaloTag from pJEx5, with the T2A sequence included in the primers:

HaloTag-fw: CAGTG GTTCA AAGTT TTTTT CTTCC ATTTC AGGTG TCGTG AGATG GAAAT CGGTA CTGGC TTTCC ATTCG.

HaloTag-rev: GCTGT CAGTT CAGGT TTCTT CATGG TGGGT GGGCC AGGAT TCTCC TCCAC GTCAC CGCAT GTTAG GAGAC TTCCT CTGCC CTCCC CGGAA CCCCC ACCGC CGGAA ATCTC GAGCG TGG.

The piggyBac transposon vector was digested with NheI-HF (NEB, R3131S) and a two-fragment Gibson assembly was carried out. This generated the vector pJEx226, where the hygromycin and HaloTag are both expressed under the EF1a promoter and separated by a T2A. pJEx226 was subsequently digested with SpeI-HF (NEB, R3133S) and MluI-HF (NEB, R3198S) to insert the RABGAP1 gene after a CAG promoter. To do this, the RABGAP1 WT and mutant sequences were PCR amplified using the following primers (5′–3′):

RABGAP1-fw: CATTT TGGCA AAGAA TTGTG TACAA GATGG ATGAC AAGGC TTCTG TTGGA AAAAT C.

RABGAP1-rev: CTGCA CCTGA GGAGT GAATT CACGC GTTAG CAAGT CTCTT TCCCT TGAAC.

## Mammalian cell culture

### HeLa cell culture

HeLa cells were cultured in Dulbecco's modified Eagle's medium (DMEM) (Sigma, D6429) supplemented with 1% MycoZap (Lonza, VZA-2012) and 10% fetal calf serum (FBS) (Merck, F7524) at 37 °C with 5% $CO_2$.

To achieve transient CRISPR knockouts, $2.5 \times 10^4$ HeLa cells stably expressing both Cas9 and HaloTag-APP-mNeonGreen were seeded per well of a 48-well plate, in a total of 170 µl media. 30 µl of 10× concentrated lentiviral particles (carrying two CRISPR guides/gene) was added to each well, and the plate was spun at 700 relative centrifugal force (RCF) for 1 h at 37 °C. After 48 h, cells were replated into a 6-well plate and incubated for 4 days. On day 6, cells were used for flow cytometry assays or immunoblotting, as described below. Depletion of proteins of interest was verified by immunoblotting. The CRISPR guide sequences (5′–3′) targeting RABGAP1 were as follows:

RABGAP1 guide 1: CAGTGCTTTAAACTACGCCA; RABGAP1 guide 2: ACTCCTGGGAGCATTTACCG.

### Flow cytometry assays

HeLa cells stably expressing HaloTag-APP-mNeonGreen (with or without RABGAP1 KO) were stained with HaloTag-JF646 ligand for 24 h. The HaloTag JF646 ligand was then removed for 24 h to monitor the processing of the stained population. Cells were then lifted with trypsin, pelleted at 500 RCF for 5 min, and washed in PBS (supplemented with 100 mg/l calcium chloride ($CaCl_2$) and 100 mg/l magnesium chloride ($MgCl_2$) ions (Gibco, 14040117). Samples were re-suspended in 500 µl ice-cold PBS and filtered using Cell-Strainer-capped 5 ml round-bottom tubes (Corning, 352235). A minimum of 30,000 cells were analysed per condition using an LSRFortessa cell analyzer (BD Biosciences), gating for mNeonGreen and HaloTag-JF646-positive cells. Data were analyzed using FlowJo software (FlowJo Software, version 10.7.1, for Mac OS X, Becton Dickinson and Company [BD] 2006–2020). Kinetic trafficking assays using RUSH were performed as previously described in Pereira et al (2023).

### Generation of lentiviral particles

Lenti-X 293T cells were used to package the pKLV guide vectors (pKLV-PU6-gRNA-BbsI-pPGK-Zeocin-mTagBFP2) and proteins for over-expression into lentiviral particles. On day 1, $0.8 \times 10^6$ Lenti-X

293T cells were seeded in one well of a six-well plate, in 2 ml media without antibiotics. On the following day, a transfection was carried out using the following protocol: For one well of cells, 1 µg 199 Gag/Pol plasmid, 0.34 µg 977 VSVG plasmid and 8.3 µl P3000 reagent (Invitrogen, L3000001) were added to 116 µl Opti-MEM (Gibco, 31985047) (master mix 1). A second master mix was made up using 120 µl Opti-MEM, 9.6 µl LTX reagent (Invitrogen, A12621) and 2.4 µl Lipofectamine 3000 (Invitrogen, L3000001) per well. 125 µl master mix 1 was added to eppendorfs containing 0.5 µg/guide DNA (1 µg total) and thoroughly mixed. 125 µl of master mix 2 was then also added and thoroughly mixed, and the solution was left to incubate for 30 min at room temperature (RT). After this, 250 µl transfection mix was added to each well with 2 ml fresh media. After 24 h, the media was removed and replaced with 3 ml fresh media. After 72 h, the viral supernatants were harvested, filtered using a 0.45 µm filter, and concentrated to 10–20× using a Lenti-X Concentrator (Takara Bio, 631232). The shRNA-containing lentivirus was produced by the Janelia Viral Tools team at HHMI Janelia Research Campus (Ashburn, VA).

### iPSC culture

Human WTC11 iPSCs with a stably integrated doxycycline-inducible NGN-2 transgene (Passage 22) (termed i3 neurons) were gifted to us by Micheal Ward (National Institutes of Health, USA). Cells were cultured on hESC-qualified Matrigel (Corning, 354277) in Essential-8 medium (Gibco, A1517001) supplemented with 10 µM Rho-associated protein kinase inhibitor Y-27632 (ROCKI) (Tocris Bioscience, 1254). 24 h after initial thawing/passaging, ROCKI was removed. Cells were passaged at approximately 80% confluency using 0.5 mM EDTA (Gibco, AM9260G). Accutase (Gibco, A1110501) was used for passaging when cell singularisation was required. All centrifugation steps were carried out at 300 RCF for 5 min at RT. Cell cultures were maintained at 37 °C in a humidified 5% $CO_2$ atmosphere.

### Differentiation of iPSCs to i3 neurons

Neuronal induction media (IM) contained the following: 48.5 ml DMEM/F12 (Gibco, 11330032); 0.5 ml N2 100x (Gibco, 17502048); 0.5 ml non-essential amino acids (NEAA) 100x (Gibco, 11140050); 0.5 ml GlutaMAX 100x (Gibco, 25030081); 2 µg/ml doxycycline in PBS (Sigma, D9891); 10 µM ROCKI. On day 0, iPSCs were re-suspended in IM and plated onto matrigel-coated plates with $5 \times 10^6$ cells per 10-cm dish. Complete media changes were carried out on days 1–2. On day 3, cells were lifted in Accutase, re-suspended in cortical neuron culture medium (CM), counted and plated onto Poly-L-Ornithine (Sigma, P3655) (PLO)-coated plates. For biochemical applications, $1.5 \times 10^6$ cells/well were plated into six-well plates in 2 ml CM. At day 3, cells were also frozen for later use, in 90% KnockOut serum replacement (Gibco, 10828-010) with 10% DMSO.

CM was made up as follows: 49 ml BrainPhys Media (Stemcell Technologies, 05790); 1 ml B27 Supplement 50x (Gibco, 17504044); 10 ng/ml BDNF (solubilised in 0.1% IgG and protease-free BSA in PBS) (PeproTech, 450-02); 10 ng/ml NT-3 (solubilised in 0.1% IgG and protease-free BSA in PBS) (PeproTech, 450-03); 1 µg/ml laminin (Gibco, 23017015); 20 mM D-glucose (Sigma-Aldrich, G7021). CM media was supplemented with 1 µg/ml doxycycline on day 3 only, as prolonged exposure to doxycycline appeared to significantly improve the health of neuronal cultures. A half-media change was carried out every 3 days for high-density cultures used

for biochemistry and every 5 days for low-density cultures used for imaging.

### Generation of stable CRISPRi KD iPSCs

Stable CRISPRi KD lines were generated in iPSCs already engineered to stably express dCas9. $0.5 \times 10^6$ cells/well were seeded in 1 ml E8 supplemented with 10 µM ROCKI in a six-well plate. 950 µl E8 with ROCKI was mixed with 50 µl 20× lentivirus (carrying 2 CRISPRi guides/target) and 2 µl polybrene. This was added to the six-well and the plate was spun at 300 RCF for 1 h at 37 °C. After 48 h, 1 µg/ml puromycin was added to the cells (resistance conferred by the CRISPRi guide plasmids) to positively select those with a KD. After 7 days, the cells were sorted via flow cytometry for those expressing mTagBFP2. The mTagBFP2 marker was present in the CRISPRi guide plasmid, enabling positive selection of cells with a CRISPRi KD. The CRISPRi guide sequences were designed using the CRISPick Broad Institute guide design tool (https://portals.broadinstitute.org/gppx/crispick/public). The following CRISPRi guides (5′–3′) were used:

RABGAP1 guide 1: AGGCG GCGGA GCCTC CGGGA;
RABGAP1 guide 2: CCCGC CGCTC GCCGT CCCGG;
RABGAP1L guide 1: CCCGG CTTCA GAGCG CGAGG;
RABGAP1L guide 2: CGGAG CGAAC GGGAC CGGCC.

### Stable overexpression of RABGAP1 in iPSCs

For complementation and overexpression experiments, either WT RABGAP1 or a RABGAP1 mutant was cloned into the piggyBac transposon vector to generate stable iPSCs overexpressing RABGAP1. For the recovery cell lines, a ratio of 5:1 of transposon:transposase was used in the RABGAP1 KD iPSCs to achieve low expression levels of RABGAP1. For the overexpression lines, a ratio of 1:1 was used. RABGAP1 KD cells were lifted with Accutase and plated at $1 \times 10^6$ cells/well in a six-well plate in 2 ml E8 supplemented with ROCKI. The cells were left to adhere for 2 h. 3 µg total DNA was added to 100 µl Opti-MEM (Gibco, 31985047) and mixed, whilst in a separate Eppendorf, 10 µl Lipofectamine Stem (Invitrogen, STEM00001) was added to 100 µl Opti-MEM. These solutions were combined and incubated at RT for 10 min before being added to the cells. After 48 h, 50 µg/ml hygromycin was added to the cells (resistance conferred by the RABGAP1 piggyBac vector). After 2 weeks, the cells were sorted via flow cytometry for HaloTag-positive cells.

### Primary rat hippocampal neuronal culture

Animal work and housing were performed in accordance with the UK Animals (Scientific Procedures) Act 1986 and the non-regulated scientific procedure described below was approved by the Animal Welfare and Ethical Review Body of the University of Cambridge. Eleven-day pregnant Sprague-Dawley rats were purchased from Charles River UK Limited. They were individually caged with access to food and water. The female rats were euthanised by CO2 inhalation followed by cervical dislocation at embryonic day 18. The embryos were used for hippocampal neuronal cultures.

The hippocampi from four embryos were dissected and kept on ice in HBSS (Merck, H2387) with 20% FBS (Merck, F7524). They were washed three times in 8 ml HBSS by inverting the 15 ml Falcon tube, letting the tissue settle into a pellet, and aspirating the solution. In total, 7 µl 12.5 mg/ml DNase (Sigma, D5025-15KU)

was added to 5 ml HBSS and used to resuspend one vial of papain dissociation powder (Lorne Labs, LK003178). The papain solution was then filter-sterilised and added to the tissue. This was incubated in a water bath at 37 °C for 7 min. After this, the solution was aspirated and the tissue was washed twice in 8 ml HBSS and 20% FBS, followed by three washes in HBSS. 7 µl 12.5 mg/ml DNase was added to 5 ml pre-prepared dissociation solution and filter-sterilised before being added to the tissue. The dissociation solution was prepared with 100 ml HBSS + 295 mg MgSO₄·7H₂O, filter-sterilised and kept at 4 °C. The tissue was then broken up using a P1000 and the subsequent solution was passed through a 40 µm cell strainer (Corning, 352340) to singularise cells. The cell solution was then centrifuged at 300 RCF for 5 min at 4 °C. A final wash in HBSS was carried out and the cell solution was centrifuged again. After this, the pellet was re-suspended in neuronal plating media and the cells were distributed into 2–3 six-well plates. Plates were coated with 0.01% PLO overnight at 37 °C, followed by a second coating in 5 µg/ml laminin for approximately 4 h at RT. The plating media was made up of 200 µl NeuroCult SM1 supplement (Stemcell Technologies, 05711) with 100 µl GlutaMAX 100x (Gibco, 35050061) in 9.7 ml NeuroCult Neuronal Plating Medium (Stemcell Technologies, 05713).

### Manipulation of RABGAP1 in primary hippocampal neuronal culture

On DIV5, a half-media change was carried out using neuronal feeding media. Feeding media was made up of 49 ml BrainPhys Media (Stemcell Technologies, 05790) and 1 ml NeuroCult SM1 supplement 50× (Stemcell Technologies, 05711). For RABGAP1 overexpression, 100 µl 20× RABGAP1-mEmerald lentivirus was added to cells on DIV5. After this, a half-media change was carried out every 4–5 days using feeding media, until the neurons reached DIV21 when they were lysed for immunoblotting. For RABGAP1 KD in primary neurons, lentivirus containing RABGAP1 shRNAs was added to the neurons on DIV5. Lentiviral GFP vectors containing 29mer shRNAs targeting rat RABGAP1 were obtained from Origene (TL706390). The RABGAP1 shRNA sequence (5′-3′) used was: GAACA GGCAT TCAGT GTTCT GGTCA AGAT.

### Live cell fluorescence microscopy

Proteins such as mCherry-RABGAP1 were transiently expressed in HeLa cells via FuGENE transfection. Here, $9 \times 10^4$ cells were seeded onto glass matrigel-coated coverslips in 6-well plates. 40 µl opti-MEM was added to 2–4 µg DNA, depending on the size of the vector. In a separate Eppendorf, 148 µl opti-MEM was added to 12 µl FuGENE 6 transfection reagent (Promega, E2691). These solutions were incubated separately for 5 min at RT, before being combined and incubated for a further 20 min. The 200 µl solution was added to one well of a six-well plate containing 4 ml fresh cDMEM. After 2 h, the media was replaced. Cells were imaged the following day using a Zeiss Elyra 7 Lattice SIM (structured illumination microscopy) microscope, equipped with two PCO.edge sCMOS version 4.2 (CL HS) cameras (PCO), solid state diode continuous wave lasers and a Plan-Apochromat 63x/1.4 Oil DIC M27 objective, at the CIMR Light Microscopy Facility.

For imaging of i3 neurons, coverslips were coated overnight with 0.01% PLO at 37 °C, followed by a second coating in 5 µg/ml laminin for approximately 4 h at RT. 200,000 cells were seeded per coverslip on day 3 of differentiation. On day 5, cells were infected with 5–10 µl of 20× lentivirus for overexpression of fluorescently

tagged proteins. Neurons were imaged on day 15–16 of differentiation, using a Zeiss LSM880 with Airyscan GaAsP array detector and Plan-Apochromat 63×/1.4 Oil DIC M27 objective at the CIMR Light Microscopy Facility.

## Biochemical assays

### Affinity chromatography of GST-tail fusions with HEK293T lysate

GST fused to the cytosolic tail of APP (either WT or a mutant) was expressed in T7 express *E. coli* (NEB, C2566H) and grown in 1 L of shaking 2xTY at 37 °C, to an $OD_{600}$ of 0.7. Expression was induced with 0.5 mM IPTG and grown overnight at 18 °C. The bacteria were harvested via centrifugation (6240 RCF, 20 min, 4 °C) and washed in 30 ml/pellet of cold PBS. The bacteria was re-suspended in lysis buffer composed of 50 mM Tris pH 7.4, 150 mM NaCl, 1 mM EDTA, 5 mM 2-mercaptoethanol, 1% triton-X100 and 1× EDTA-free protease inhibitor cocktail (Roche, 11836170001). The bacteria was lysed for 30 min on ice, before sonication (3 × 1 min sonications) and clarification via ultracentrifugation (126,100 RCF, 35 min, 4 °C). The supernatant was flash-frozen in aliquots equal to 500 ml bacteria.

Two confluent 15 cm dishes of HEK293T were used per condition for affinity chromatography. To prepare the mammalian cell lysate, HEK293T cells were collected in PBS, centrifuged (550 RCF, 5 min, RT), and washed once in PBS. Here, all lysates were pooled together to ensure homogeneity. Cells were then incubated in lysis buffer (detailed above) on ice for 30 min, before clarification via ultracentrifugation (17,000 RCF, 20 min, 4 °C). The supernatant was flash frozen in aliquots equal to two 15-cm plates.

For affinity chromatography, 50 µl glutathione-Sepharose 4B beads slurry (Merck, GE17-0756-04) was used per sample. Beads were washed three times in lysis buffer before use. 100 µl 50% bead slurry was added to each GST-APP tail sample and incubated on a roller at 4 °C for 2 h, to load the GST-APP protein onto the glutathione beads. HEK293T lysate was also pre-cleared with empty glutathione beads (50 µl beads/sample) on a roller at 4 °C for 2 h. The GST-loaded beads were washed once with cold lysis buffer, once in lysis buffer supplemented with 500 mM NaCl, and once finally in lysis buffer. Beads were incubated with pre-cleared HEK293T cell lysate overnight on a roller at 4 °C. On the following day, the beads were washed twice in lysis buffer before being transferred to 0.8 ml centrifuge columns (Pierce, 89869B) for two further washes. The columns were then brought to RT and five sequential elution steps were carried out using 100 µl elution buffer (1.5 M NaCl in lysis buffer) per column. The eluate from each column was pooled and concentrated in a 3 KD Amicon column (Merck, UFC5003) to just under 100 µl. After elution from the Amicon columns, each sample was weighed and equilibrated to precisely 100 µl using elution buffer. Samples were analysed via either mass spectrometry or immunoblotting. For immunoblotting, 4× Laemmli buffer (BioRad, 1610747) supplemented with 100 mM DTT (final DTT concentration 25 mM) was added to each sample and incubated for 10 min at 95 °C. These samples were stored at −80 °C for use in immunoblotting.

### Mass spectrometry analysis of eluates from affinity chromatography

The following protocol was adapted from Cattin-Ortolá et al (2024). The eluates were separated by SDS-PAGE and stained with InstantBlue Coomassie (Abcam, ab119211), before being cut into multiple fragments and transferred to a 96-well plate. The gel slices were destained with 50% (v/v) of acetonitrile and 50 mM ammonium bicarbonate, reduced with 10 mM DTT, and alkylated with 55 mM iodoacetamide. After this, trypsin (Promega, UK) was used to digest proteins overnight at 37 °C. The resulting peptides were then extracted using 2% (v/v) formic acid and 2% (v/v) acetonitrile. Nanoscale capillary liquid chromatography (LC)–MS/MS was then performed to separate the digests, using an Ultimate U3000 high-performance LC (HPLC) (Thermo Fisher Scientific, Dionex, San Jose, USA) and a flow rate of 300 nl/min. A C18 Acclaim PepMap100 5 µm, 100 µm × 20 mm nanoViper (Thermo Fisher Scientific, Dionex, San Jose, USA) was used to trap peptides before separation using a C18 BEH130 1.7 µm, 75 µm × 250 mm analytical ultrahigh-performance LC column (Waters, UK). Peptide elution was performed using a 60 min gradient of acetonitrile (2–80%). The analytical column outlet was directly interfaced with a quadrupole Orbitrap mass spectrometer (Q- Exactive HFX, Thermo Fisher Scientific), via a nanoflow electrospray ionization source. The data was collected in data-dependent mode using a top 10 method. Here, ions with a precursor charge state of 1+ were excluded. High-resolution full scans ($R = 60,000$; $m/z$, 300–1800) were recorded in the Orbitrap, followed by higher energy collision dissociation (26% normalised collision energy) of the 10 most intense MS peaks. The fragment ion spectra were recorded at a resolution of 15,000, with a dynamic exclusion window of 20 secs. Raw data files were processed using Proteome Discoverer v2.1 (Thermo Fisher Scientific) and searched against a human protein database (UniProtKB) using the Mascot search engine programme. Database search parameters were set with a precursor tolerance of 10 ppm and a fragment ion mass tolerance of 0.2 Da. One missed enzyme cleavage was considered acceptable. Variable modifications for oxidised methionine, carbamidomethyl cysteine, pyroglutamic acid, phosphorylated serine, threonine, and tyrosine were also included. MS/MS data was then validated using Scaffold. To analyse mass spectral intensities, all raw files underwent processing using MaxQuant v1.5.5.1 with standard settings, utilising the Andromeda search engine integrated within the MaxQuant software suite (Cox and Mann, 2008). The search was conducted against the UniProt database. Enzyme specificity was set to Trypsin/P for both endoproteases, allowing for up to two missed cleavages per peptide. Carbamidomethylation of cysteines was designated as a fixed modification, while oxidised methionine and protein N-acetylation were considered variable modifications. The search parameters included an initial mass tolerance of 6 ppm for precursor ions and 0.5 Da for MS/MS spectra. A false discovery rate of 1% was applied at both the peptide and protein levels. Statistical analysis was performed using the Perseus module of MaxQuant (Tyanova et al, 2016). Peptides associated with known contaminants and reverse hits were excluded from the analysis. Only protein groups identified with at least two peptides, one of which was unique, and with two quantitation events were considered for data analysis. Each protein had to be detected in at least two out of the three replicates. Missing values were inputed using Perseus' default settings, simulating noise.

### Mass spectrometry data analysis

A PANTHER GO analysis of mass spectrometry hits from affinity chromatography using GST fused to the APP tail was performed using the PANTHER 18.0 database (https://pantherdb.org/tools/

compareToRefList.jsp) and a PANTHER overrepresentation test (Fisher's exact). The GO cellular component complete annotation dataset was selected. Hits with a fold enrichment of ≥twofold were included in the analysis, totaling 101 proteins. GO categories with a FDR $P$ value < 0.05 were included.

The BioGRID database was used to map protein-protein interactions of hits with $P \le 0.01$. Positive interactions between proteins were mapped as connected nodes. All displayed proteins interacted with the APP tail; however, for the sake of clarity, these were not displayed as connected nodes. This analysis was performed in Python 3.9 using the BioGRID API, with the networks generated using NetworkX and formatted for publication in Adobe Illustrator.

### Preparation of whole-cell lysates for immunoblotting

The following protocol was used to prepare i3 neuron, primary rat neuron and HeLa cell lysates. For i3 neurons, $1.5 \times 10^6$ cells were seeded per well of a six-well plate. For primary rat neurons, 4 rat hippocampi were distributed evenly between 3 and 4 six-well plates. Where specified, neurons were treated with either 25 μM DAPT for 24 h or 10 μM MG132 for 1 h before cell lysis. i3 neurons were lysed on day 15. Primary rat neurons were lysed on DIV21. In all, 5 ml lysis buffer was made up using 5 ml RIPA buffer with 5 μl 1 M PMSF and 1× EDTA-free protease inhibitor cocktail (Roche, 11836170001). Cells were washed with 1 ml cold PBS before 250 μl lysis buffer was added to each well. Cells were subsequently scraped into 1.5 ml low protein binding Eppendorfs (Sarstedt, 72.706.600) on ice. 1 μl benzonase was added to each Eppendorf, and samples were incubated on a rotating wheel at 4 °C for 1 h. Lysates were then clarified via centrifugation (17,000 RCF, 1 h, 4 °C). The supernatants were transferred to new low-binding Eppendorfs and the protein content of each sample was quantified using a Bradford assay. Immunoblotting samples were made up with 6x Laemmli buffer supplemented with 75 mM DTT (final DTT concentration 12.5 mM), with 40–60 μg protein per sample. Samples were then incubated at 95 °C for 10 min, before storage at −80 °C for subsequent immunoblotting analysis.

### Immunoblotting

Denatured cell extracts were resolved on homemade 10% Tris–Glycine acrylamide gels. Proteins were transferred to a polyvinylidene difluoride (PVDF) membrane using a wet transfer protocol (110 V for 2 h) and blocked with 5% BSA in PBS with 0.1% Tween-20 (PBS-T) for 1 h. Membranes were incubated with the appropriate primary antibody overnight on a rocker at 4 °C. The following day, membranes were washed three times with PBS-T and incubated with a 1:5000 HRP-conjugated secondary antibody (Abcam, ab205718), diluted in 5% BSA in PBS-T, for 1 h at RT. The membranes were visualised using either Clarity (Bio-Rad, 1705061) or WesternBright Sirius (Advansta, K-12043-D10) enhanced chemiluminescence solutions and a ChemiDoc Imaging System equipped with the ImageLab software (Bio-Rad Laboratories). After initial visualisation of the protein of interest (usually APP or RABGAP1), membranes were subsequently incubated with β-tubulin as a total protein control during quantification. β-tubulin was visualised using a mouse Alexa Fluor-647 conjugated secondary antibody (Invitrogen, A32728). Protein bands in immunoblots were quantified using Image Lab software version 6.1 (BioRad).

### Antibodies

The following primary antibodies were used for immunoblotting in this study: APP/C99 (D54D2) (Cell Signalling Technology, 8243, 1:2000), APP/AICD (Invitrogen, 51-2700, 1:2000), APP/C99 (82E1) (Stratech Scientific, 10323), RABGAP1 (Abcam, ab153992, 1:2000), β-tubulin (ProteinTech, 66240-1-Ig, 1:100,000).

### Isothermal titration calorimetry

The His-tagged PTB domain of RABGAP1 was purified from 8 L of T7 express *E. coli*, as previously described. Cells were re-suspended in 200 ml of lysis buffer, consisting of 350 mM NaCl, 10 mM Tris (pH 7.4), 1:10,000 β-mercaptoethanol and supplemented with 100 μl 4-benzene sulfonyl fluoride hydrochloride (AEBSF), $MnCl_2$ and deoxyribonuclease I. Cells were lysed using a mechanical cell disruptor (Constant Systems), before ultracentrifugation at 30,000 RCF for 40 min. A Ni-NTA bead column was equilibrated in lysis buffer, before incubation with the His-PTB protein lysate on a roller for 40 min at 4 °C. The Ni-NTA bead resin was washed with 400 ml lysis buffer supplemented with 10 mM imidazole pH 8, before elution in lysis buffer supplemented with 300 mM imidazole. Eluates were concentrated using a 3 kD Amicon column for gel filtration. Gel filtration was performed using a Hiload 26 60 Superdex 200 pg column (Cytiva/GE Healthcare) to buffer exchange His-PTB into ITC buffer (100 mM Tris (pH 7.4), 300 mM NaCl and 1 mM tris(2-carboxyethyl)phosphine (TCEP)).

After gel filtration, the sample was concentrated to approximately 100 μM in a 3 kD Amicon column for ITC. ITC experiments were performed using a Nano ITC machine (TA Instruments). The APP peptides (either WT or mutant) were dissolved in ITC buffer. Peptides were titrated into the RABGAP1 PTB domain in 2.4 μl injections at 20 °C. For the WT APP peptide that displayed measurable binding to the RABGAP1 PTB domain, three independent runs were carried out that demonstrated clear saturation of binding. These were used to calculate the mean $K_D$ of the reaction and the stoichiometry ($n$).

### RNA extraction and qPCR

RNA was extracted from both HeLa cells and i3 neurons using an RNeasy mini kit (Qiagen, 74104) and quantified using a NanoDrop One (Thermo Fisher Scientific). In total, 1 μg of total RNA was subsequently used for cDNA synthesis, along with an oligo (dT) 16/random hexamers mix and a High-Capacity RNA-to-cDNA Kit (Applied Biosystems, 4387406). cDNA samples were diluted 10-fold and qPCR reactions were set up using 1 μl of 1:10 cDNA, specific oligos spanning 2 exons and the PowerUp SYBR Green Master Mix (Applied Biosystems, A25741). All qPCR reactions were performed in three biological replicates, with three technical replicates each time, using the CFX96 Touch Real-Time PCR Detection System (Bio-Rad). TBP levels were used as a control. Data was analysed using the $2^{-\Delta\Delta CT}$ method for normalised expression (Livak and Schmittgen, 2001). The following oligo sequences were used (5′–3′):

APP-fw: GACAG ACAGC ACACC CTAAA;
APP-rev: CACAC GGAGG TGTGT CATAA;
TBP-fw: GAGAG TTCTG GGATT GTACC G;
TBP-rev: ATCCT CATGA TTACC GCAGC.

### AlphaFold3 modelling

Protein-protein interactions were modelled in AlphaFold3, accessed at https://alphafoldserver.com. For each model, five output

models were generated and compared using UCSF ChimeraX (version 1.8). The Predicted Aligned Error (PAE) plots were generated using Python 3.9 to visualise the predicted uncertainty in the relative positions of residues in the protein models.

### Statistical analysis of all other figures

Statistical data are demonstrated as mean ± SEM and a minimum of $N = 3$ independent biological repeats was used in each experiment. The statistical test used to determine significance is described in each figure legend and statistical significance was considered when $P < 0.05$. When only two samples were compared, a Student's $t$ test was used. When comparing multiple conditions, an analysis of variance (ANOVA) test was initially conducted using Ordinary Least Squares (OLS) regression. This was followed by a Multiple Comparison of Means Tukey's Honest Significant Difference (HSD) test, with a family-wise error rate (FWER) of 0.05. All data was plotted and analysed using Python 3.9 in either PyCharm CE (JetBrains) or Jupyter Notebook, using the statsmodels.stats.multicomp or scipy.stats packages. The p-values are labelled as follows: $*P \leq 0.05$; $**P \leq 0.01$; $***P \leq 0.001$ and $****P \leq 0.0001$. Differences were considered statistically significant at $P \leq 0.05$.

## Data availability

This study includes no data deposited in external repositories. The mass spectrometry dataset in Fig. 1 is available in Dataset EV1.

The source data of this paper are collected in the following database record: biostudies:S-SCDT-10_1038-S44318-025-00530-0.

## Peer review information

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

## Acknowledgements

We thank Ralph Boettcher (Max Planck Institute of Biochemistry, Germany) for the generous gift of DNA plasmids. We also thank Luke Levis (Janelia Research Campus, USA) for generously providing all JF HaloTag ligands. This study was supported by the Cambridge Institute for Medical Research Flow Cytometry Core Facility and Microscopy Facility. In particular, we wish to thank Reiner Schulte and Gabriela Grondys-Kotarba for their advice and support in flow cytometry and cell sorting. We thank Laura Pellegrini for reading the manuscript. We would like to thank NIAID Visual & Medical Arts for the use of their neuron image in our graphical abstract (07/10/2024), sourced from the NIAID NIH BIOART collection (bioart.niaid.nih.gov/bioart/197). DC Gershlick is funded by a Sir Henry Dale Fellowship awarded to DC Gershlick from the Wellcome Trust/Royal Society (Grant 210481) as well as a Royal Society Research Grant, United Kingdom (grant no.: RGS/R2/202082) and a UKRI BBSRC research grant (BB/W005905/1). J Eden is funded by the BBSRC Doctoral Training Partnership, United Kingdom (grant no.: BB/M011194/1). D Owen and J Kaufman are funded by a Wellcome Trust PRF awarded to D Owen (207455/Z/17/Z) and a Wellcome Discovery Award (227915/Z/23/Z). B Nieuwenhuis is funded by the Medical Research Council (MR/V002694/1). J Lippincott-Schwartz and L Benedetti are funded by the Howard Hughes Medical Institute through Janelia Research Campus, as well as the visiting scientist program for L Benedetti. For the purpose of open access, the author has applied a Creative Commons Attribution (CC BY) license to any Author Accepted Manuscript version arising.

## Author contributions

**Jessica Eden**: Conceptualization; Data curation; Formal analysis; Writing—original draft; Writing—review and editing. **Jonathan G G Kaufman**: Conceptualization; Data curation; Formal analysis. **Conceição Pereira**: Data curation; Formal analysis; Investigation; Methodology. **Eleanor Fox**: Resources; Data curation; Investigation; Methodology. **Jerome Cattin-Ortolá**: Resources; Data curation; Formal analysis; Methodology. **Lorena Benedetti**: Resources; Methodology. **Bart Nieuwenhuis**: Resources; Supervision; Methodology. **David J Owen**: Resources; Methodology; Writing—review and editing. **Jennifer Lippincott-Schwartz**: Conceptualization; Resources; Software; Formal analysis; Supervision; Funding acquisition; Investigation; Methodology; Writing—original draft; Writing—review and editing. **Sean Munro**: Resources; Data curation; Investigation; Methodology; Writing—review and editing. **David C Gershlick**: Conceptualization; Resources; Data curation; Software; Formal analysis; Supervision; Funding acquisition; Investigation; Writing—original draft; Writing—review and editing.

Source data underlying figure panels in this paper may have individual authorship assigned. Where available, figure panel/source data authorship is listed in the following database record: biostudies:S-SCDT-10_1038-S44318-025-00530-0.

## Disclosure and competing interests statement

The authors declare no competing interests.

