## [Peer Review File · The EMBO Journal]

RABGAP1 is a sensor that facilitates the sorting and processing of amyloid precursor protein

Jessica Eden, Jonathan Kaufman, Conceição Pereira, Eleanor Fox, Jerome Cattin-Ortola, Lorena Benedetti, Bart Nieuwenhuis, David Owen, Jennifer Lippincott-Schwartz, Sean Munro, and David Gershlick

Corresponding author(s): David Gershlick (dg553@cam.ac.uk)

Review Timeline:

Submission Date:	18th Sep 24
Editorial Decision:	2nd Oct 24
Appeal Received:	8th Nov 24
Editorial Decision:	2nd Jan 25
Revision Received:	1st May 25
Editorial Decision:	24th Jun 25
Revision Received:	14th Jul 25
Accepted:	21st Jul 25

Editor: William Teale

Transaction Report:

Dear David,

Thank you for submitting your study, "RABGAP1 acts as a sensor to facilitate sorting and processing of amyloid precursor protein" (EMBOJ-2024-119071), to EMBO Journal. I have now read and discussed your manuscript with my editorial colleagues. However, I am sorry to say that we have concluded that we cannot offer publication at the EMBO Journal.

We appreciate that the work you present is interesting and relevant. Using a pull-down of GST fused to the APP tail, you identify a list of candidate interactors in which endosomal trafficking proteins are prominent. One of these, RABGAP1, is chosen for further study and is shown to interact with the APP YENTPY motif via its PTB domain. This interaction is required for amyloidogenic APP cleavage in neurons, with RABGAP1 knock-down cells showing less C99 and the overexpressor more. Thus you show that a direct interaction between RABGAP1 and APP is required for amyloid-generating peptide protein processing.

However, the editorial team had two concerns: firstly on mechanistic depth, and secondly on physiological relevance. My opinion is that you would either need to show at which point RABGAP1 interacts with the APP processing machinery, or whether RABGAP1 activity affects the generation of pathogenic amyloid fragments in vivo. Without this we are not convinced that your study would be a strong candidate for publication in a broad general journal like The EMBO Journal.

Just in case you need to publish this quickly, I shared the manuscript with other EMBO Press titles. Life Science Alliance indicated that they would be happy to send out for full review.

I am really sorry I can't be more supportive, and wish you well with this manuscript.

Best wishes,

William

William Teale, PhD
Editor
The EMBO Journal
w.teale@embojournal.org

** As a service to authors, EMBO Press provides authors with the possibility to transfer a manuscript that one journal cannot offer to publish to another EMBO publication or the open access journal Life Science Alliance launched in partnership between EMBO Press, Rockefeller University Press and Cold Spring Harbor Laboratory Press. The full manuscript and if applicable, reviewers' reports, are automatically sent to the receiving journal to allow for fast handling and a prompt decision on your manuscript. For more details of this service, and to transfer your manuscript please click on Link Not Available. **

Dear William,

Thank you for your email and your thoughtful comments. We agree this manuscript would be strengthened by identifying at what point in the endomembrane system APP interacts with the RABGAP1 and thank you for highlighting this avenue for us. Accordingly we have undertaken a set of experiments to address this. Using lattice-SIM super-resolved live-cell imaging coupled with advanced image quantification pipelines has allowed us to make a series of insights 1) the major site of RABGAP1 is on RAB11 positive endosomes 2) the major site of processing is on this endosomal sub-population 3) this is distinct in both cases from RAB5 endosomes. Please find the modified figure attached for your consideration. We feel these observations make a step change to the quality of the manuscript. It is well accepted that APP processing happens in an endosomal sub-population but demonstrating that it is RAB11 specifically will be of importance for the field for several key reasons, not least access for targeted drug delivery. Additionally, we have bolstered the mechanistic underpinnings of our initial data to not only demonstrate that RABGAP1 is a key component, and how it is a key component, but additionally where it acts.

We hope that you agree with us that these findings strengthen the manuscript, and kindly request that the journal reconsider the initial decision. We are very happy to answer any clarifying questions or comments and we wish to reiterate that we genuinely appreciate the open and thoughtful communication from your end.

Best wishes,
David Gershlick

Dear David,

Thank you again for the submission of your manuscript entitled "RABGAP1 acts as a sensor to facilitate sorting and processing of amyloid precursor protein" and for your patience during the review process. We have now received the reports from the referees, which I copy below.

Your work identifies RabGAP1 as an interactor of APP, and shows that its function must remain intact for proper processing to take place. As you can see from their comments, the referees identify the research question you address as being both timely and important. However, referee #3 raises important and relevant concerns over whether it is the APP-RabGAP1 interaction itself that is required for correct protein processing. Whilst acknowledging the important control of a RabGAP1 version that does not bind APP, I see that potential contributions from general trafficking defects here are indeed not formally excluded. These and other concerns will require your attention before your manuscript can be published in The EMBO Journal.

Based on the overall interest expressed in the reports, however, I would like to invite you to address the comments of all referees in a revised version of the manuscript. I should add that it is The EMBO Journal policy to allow only a single major round of revision and that it is therefore important to resolve the main concerns at this stage. I believe the concerns of the referees are reasonable and addressable, but please contact me if you have any questions, need further input on the referee comments or if you anticipate any problems in addressing any of their points. I think a Zoom call would be useful to identify key experiments that will need to be included in this revision, could you let me know when would be most convenient? Please, follow the instructions below when preparing your manuscript for resubmission.

I would also like to point out that as a matter of policy, competing manuscripts published during this period will not be taken into consideration in our assessment of the novelty presented by your study ("scooping" protection). We have extended this 'scooping protection policy' beyond the usual 3 month revision timeline to cover the period required for a full revision to address the essential experimental issues. Please contact me if you see a paper with related content published elsewhere to discuss the appropriate course of action.

Again, please contact me at any time during revision if you need any help or have further questions.

Thank you very much again for the opportunity to consider your work for publication. I look forward to your revision.

Best regards,

William

William Teale, Ph.D.
Editor
The EMBO Journal

When submitting your revised manuscript, please carefully review the instructions below and include the following items:

- 1) a .docx formatted version of the manuscript text (including legends for main figures, EV figures and tables). Please make sure that the changes are highlighted to be clearly visible.
- 2) individual production quality figure files as .eps, .tif, .jpg (one file per figure).
- 3) a .docx formatted letter INCLUDING the reviewers' reports and your detailed point-by-point response to their comments. As part of the EMBO Press transparent editorial process, the point-by-point response is part of the Review Process File (RPF), which will be published alongside your paper.
- 4) a complete author checklist, which you can download from our author guidelines ([https://wol-prod-cdn.literatumonline.com/pb-assets/embo-site/Author Checklist%20-%20EMBO%20J-1561436015657.xlsx](https://wol-prod-cdn.literatumonline.com/pb-assets/embo-site/Author%20Checklist%20-%20EMBO%20J-1561436015657.xlsx)). Please insert information in the checklist that is also reflected in the manuscript. The completed author checklist will also be part of the RPF.
- 5) Please note that all corresponding authors are required to supply an ORCID ID for their name upon submission of a revised manuscript.

6) We require a 'Data Availability' section after the Materials and Methods. Before submitting your revision, primary datasets produced in this study need to be deposited in an appropriate public database, and the accession numbers and database listed under 'Data Availability'. Please remember to provide a reviewer password if the datasets are not yet public (see <https://www.embopress.org/page/journal/14602075/authorguide#datadeposition>). If no data deposition in external databases is needed for this paper, please then state in this section: This study includes no data deposited in external repositories. Note that the Data Availability Section is restricted to new primary data that are part of this study.

Note - All links should resolve to a page where the data can be accessed.

8) For data quantification: please specify the name of the statistical test used to generate error bars and P values, the number (n) of independent experiments (specify technical or biological replicates) underlying each data point and the test used to calculate p-values in each figure legend. The figure legends should contain a basic description of n, P and the test applied. Graphs must include a description of the bars and the error bars (s.d., s.e.m.).

9) We would also encourage you to include the source data for figure panels that show essential data. Numerical data can be provided as individual .xls or .csv files (including a tab describing the data). For 'blots' or microscopy, uncropped images should be submitted (using a zip archive or a single pdf per main figure if multiple images need to be supplied for one panel). Additional information on source data and instruction on how to label the files are available at .

10) We replaced Supplementary Information with Expanded View (EV) Figures and Tables that are collapsible/expandable online (see examples in <https://www.embopress.org/doi/10.15252/emboj.201695874>). A maximum of 5 EV Figures can be typeset. EV Figures should be cited as 'Figure EV1, Figure EV2' etc. in the text and their respective legends should be included in the main text after the legends of regular figures.

12) Our journal encourages inclusion of *data citations in the reference list* to directly cite datasets that were re-used and obtained from public databases. Data citations in the article text are distinct from normal bibliographical citations and should directly link to the database records from which the data can be accessed. In the main text, data citations are formatted as follows: "Data ref: Smith et al, 2001" or "Data ref: NCBI Sequence Read Archive PRJNA342805, 2017". In the Reference list, data citations must be labeled with "[DATASET]". A data reference must provide the database name, accession number/identifiers and a resolvable link to the landing page from which the data can be accessed at the end of the reference. Further instructions are available at .

13) In order to increase the reproducibility and reach of your work, The EMBO Journal includes a table of reagents that were used in the study. Please provide this along with your revisions.

Further instructions for preparing your revised manuscript:

We realize that it is difficult to revise to a specific deadline. In the interest of protecting the conceptual advance provided by the work, we recommend a revision within 3 months (2nd Apr 2025). Please discuss the revision progress ahead of this time with the editor if you require more time to complete the revisions. Use the link below to submit your revision:

Referee #1:

In a comprehensive, unbiased mass spectrometry screen, Eden et al. identified several novel interacting partners of the Amyloid Precursor Protein (APP) cytosolic tail, including PDLIM7 and RABGAP1. Using multiple in vitro approaches and diverse cellular models (including iPSC-derived i3 neurons and primary rat hippocampal neurons), the authors convincingly show that RABGAP1 directly binds to the APP tail, with this interaction depending on both the YENPTY motif in APP and RABGAP1 activity.

Their findings suggest a regulatory role for RABGAP1 in APP trafficking and/or processing, as both depletion (KO or KD) and overexpression of RABGAP1 significantly altered levels of full-length APP and its C-terminal fragment (presumed to be APP-C99). Importantly, while the data indicate changes in APP processing to APP-C99, the analysis and identification of APP-C99 require revision to strengthen these conclusions (see comment 1 below). Additionally, the analysis of APP-CTF levels following gamma-secretase inhibition needs reassessment, as the DAPT concentration used (25µM) is excessive. Interestingly, expression of wild-type RABGAP1, but not a GAP-deficient mutant, restored APP CTF (C99) levels in RABGAP1 knockdown neurons to wild-type levels. If these findings are validated by the suggested complementary analyses (comment 2), they will provide compelling evidence that RABGAP1 regulates APP trafficking and may add key information about the alterations in the cleavage mechanism.

In the discussion, the authors propose that RABGAP1's GAP activity inactivates RAB11A, helping maintain APP within early endosomes where proteolytic processing occurs. They hypothesize that RABGAP1 deficiency would increase membrane-associated RAB11A, causing APP accumulation in recycling endosomes where processing is less efficient. While this model provides a plausible explanation for their observations, additional experimental validation would strengthen these conclusions.

Major points:

1. The analysis of endogenous APP processing in RABGAP1 knockdown i3 neurons reveals decreased levels of both 110kDa (full-length APP) and 16kDa (CTF) bands. While the authors currently compare these changes to wild-type levels individually, they should instead calculate the product/precursor ratio (APP CTFs/full-length APP), a standardized approach that better reflects APP processing efficiency. This analysis should be applied consistently across the report. Additionally, given the importance of the proper identification of APP-C99 (and not other APP CTFs) for the conclusions, the authors should employ the 82E1 antibody, which specifically recognizes C99 (and amyloid-beta peptides).
2. The current analyses would benefit from complementary measurements: i) soluble APP ectodomain fragments, and ii) total secreted amyloid-beta peptides should be considered. These data would enable a more comprehensive understanding of RABGAP1's role in APP processing.
3. The authors' use of 25 μ M DAPT for 24 hours before analysis is problematic, as this concentration far exceeds what is necessary and may affect other aspartyl proteases (including BACE1). Given that DAPT's IC50 is in the low nanomolar range, these experiments should be repeated using 2.5 μ M DAPT, which is sufficient for complete gamma-secretase inhibition. Furthermore, it's important to note that gamma-secretase inhibition can promote alpha-secretase-mediated cleavage of APP-C99 to APP-C83.
4. The ITC data indicate that the wild-type APP tail binds to RABGAP1's PTB domain with a KD of approximately 61 μ M and a 1:1 stoichiometry. The authors should address in their discussion whether this relatively weak affinity (micromolar range) is physiologically relevant for cellular interactions.

Minor comments:

- A. Are HEK cells constantly treated with Mycozap, and why?
- B. What is the parental cell line for the i3 neurons?
- C. Figure 4: L-M has an $n = 3$. Considering the variability, the authors may want to increase the number of observations to be sure about the significance.
- D. Figure 5: Please add the colocalization analysis for panel A.

Referee #2:

The manuscript by Eden et al from the Gershlick lab describes the identification of RABGAP1 as a new interaction partner of the APP C-terminus. This novel interaction regulates the trafficking and processing of APP, which the authors explore comprehensively in both HeLa cells and i-neurons. The inclusion of neuronal models significantly enhances the physiological relevance of the study.

Overall, this work is of high quality, presenting well-designed experiments that effectively support the conclusions. The manuscript is well-written and straightforward to follow. My comments are primarily technical in nature and involve requests for clarification and some additional data, which I believe can be readily addressed during a revision:

- 1- In Figure 2: why does the FFE689-Mutant bind stronger to RABGAP1 and lose its interaction with SNX17? This is not commented on by the authors. Other mutants, such as the Y682 mutant, abolishes binding to both, RABGAP1 and SNX17. It might be good to comment on this in the main text to help the reader better grasp the story.
- 2- Which Rab GTPase(s) act downstream of RABGAP1 to mediate its effects on APP trafficking? Given that the GAP-deficient RABGAP1 mutant fails to rescue the knockdown phenotype, it is likely that one or more Rab GTPases are involved. Any effort to identify or clarify this would greatly strengthen the mechanistic understanding provided by the study.
- 3- What is the identity of the compartment where RABGAP1 colocalizes with APP? Including experiments with specific organelle markers and quantifying the co-localization would improve the clarity and robustness of this finding.
- 4- How does the localization of the double-tag APP in neurons compare to the localization of endogenous APP?
- 5- Figure 5F&G: I might have misunderstood this, but why does RABGAP1 knockdown reduce the levels of C99, while at the same time not affecting the amounts of full-length APP. In Figure 4, the KO of RABGAP1 increased full length APP as a sign of reduced processing. What explanations do the authors have for these findings.

Referee #3:

In this work, the authors add another facet to the already rich information about the proteolytic processing of amyloid precursor protein (APP) and the still somewhat tenuous connection to the intracellular trafficking of APP. First, the authors use the short tail peptide of APP to carry out a proteomic screen for interactors, which identifies alongside known binding partners, RabGAP1 and some additional proteins. The authors then focus on RABGAP1. First, they carry out a biochemical characterization of the interaction using mutation scanning of the APP tail, and ITC, showing that RABGAP1 binds to the YENPTY motif of the APP tail, albeit with a rather low affinity (61 μ M). They then perturb the expression of RABGAP1 in non-neuronal and neuronal cells to show that such perturbation is associated with a perturbation of APP trafficking and of proteolytic processing of APP, most notably of a reduction of the amyloidogenic APP precursor C99 upon RABGAP1 knockdown in neurons obtained by differentiation from iPSCs.

This is a rather complicated manuscript in which many different approaches (not all of them conclusive, with quite a few loose ends) were chosen to support the main conclusions. Even after reading the MS several times I am not entirely sure I have understood all of the details as the MS is difficult to follow for someone not intricately familiar with the sophisticated toolkit available for APP. In several cases there is insufficient explanation of the experiments to assess whether they are indeed conclusive (examples are given below).

In my opinion, the main issue is whether the evidence suffices to conclude that the observed perturbation of APP trafficking and processing is indeed due to a direct interaction between the APP tail and RABGAP1, or else, whether the perturbation is indirect, i.e. caused by the perturbation of trafficking that is well known to occur upon RABGAP1 knockout. Deleting RABGAP1 is known to result in multiple cellular phenotypes that are supposed to be caused by impaired GTP-cleavage of Rab6 (or Rab11), known regulators of Golgi-endosome trafficking. In fact, the observation that the GTPase activating function is required for the APP effects supports the second view: As far as I can see, there is no evidence showing that the interaction between the two proteins is functionally required for the observed effects on APP processing (which is acknowledged by the authors). Thus, the manuscript largely confirms (and maybe extends) the notion that perturbation of post-Golgi trafficking (via specific Rabs) affects APP trafficking and processing, and it is unclear whether there is any specific additional function of RABGAP1 as insinuated.

Specific points:

a) Fig. 4A-C: The authors compare fluorescence intensities between different cell lines (wt, KO) but there is no standardization, e.g. IC for an endogenous protein. In Fig. B it appears that for some reason more cells were used for the KO than for the wt control (the tubulin band is quite a bit stronger), showing that calibration is needed to unequivocally document the alleged increase in expression level. Generally, the characterization of the RABGAP1 and APP positive structures is insufficient - what are these dots? Did the authors quantify the degree of colocalization?

b) Fig. 4 H-M. Here the authors quantify colocalization of APP with Rab11 and Rab5 respectively. I have several problems with these experiments. First, all data are based on overexpressed and tagged proteins and thus do not represent the native state. Second, even when pushing the expression levels the degree of colocalization appears to be small (cross-correlation coefficients between 0.1 and 0.15), questioning the significance of the overall rather small changes. In addition, there is no information about cell-to-cell variability, about transfection efficiency, about expression levels in the transient expression experiments that I could find in the methods section. How was the statistics done? It makes no sense to calculate significances with $n=3$ (!). Also, were these independent experiments, or just different cells from the same transfection that were analysed?

c) The supplementary videos lack a description (imaging conditions, frame rate, labeling etc.).

Response to reviewers comments

We thank the reviewers for their constructive feedback on our manuscript. In response, we have conducted additional experiments and made substantial revisions to the text. Below, we provide a detailed, point-by-point response to each comment. We believe that these suggestions have significantly enhanced the clarity of the manuscript and strengthened our proposed model.

Reviewer 1:

In a comprehensive, unbiased mass spectrometry screen, Eden et al. identified several novel interacting partners of the Amyloid Precursor Protein (APP) cytosolic tail, including PDLIM7 and RABGAP1. Using multiple *in vitro* approaches and diverse cellular models (including iPSC-derived i3 neurons and primary rat hippocampal neurons), the authors convincingly show that RABGAP1 directly binds to the APP tail, with this interaction depending on both the YENPTY motif in APP and RABGAP1 activity.

Their findings suggest a regulatory role for RABGAP1 in APP trafficking and/or processing, as both depletion (KO or KD) and overexpression of RABGAP1 significantly altered levels of full-length APP and its C-terminal fragment (presumed to be APP-C99). Importantly, while the data indicate changes in APP processing to APP-C99, the analysis and identification of APP-C99 require revision to strengthen these conclusions (see comment 1 below).

Comment 1:

1. The analysis of endogenous APP processing in RABGAP1 knockdown i3 neurons reveals decreased levels of both 110kDa (full-length APP) and 16kDa (CTF) bands. While the authors currently compare these changes to wild-type levels individually, they should instead calculate the product/precursor ratio (APP CTFs/full-length APP), a standardized approach that better reflects APP processing efficiency. This analysis should be applied consistently across the report. Additionally, given the importance of the proper identification of APP-C99 (and not other APP CTFs) for the conclusions, the authors should employ the 82E1 antibody, which specifically recognizes C99 (and amyloid-beta peptides).

To address comment 1 by the first reviewer, we have repeated immunoblotting of WT and RABGAP1 KD i3 neurons using the 82E1 antibody, to further validate that the peptide we are analysing is indeed C99. This is included as Supplemental Figure S2. As shown here, we see a consistent decrease in the abundance of endogenous C99, detected by the 82E1 antibody, in the RABGAP1 KD neurons compared to WT. We thank the reviewer for this suggestion as we feel that it has strengthened the manuscript.

In Figure 5, as suggested by the reviewer, we have included an additional analysis to show the ratio of C99 over full length APP, where we see a significant decrease in the ratio in RABGAP1 KD neurons compared to WT (Figure 5I).

We also wish to clarify that full length APP levels are unchanged in RABGAP1 KD i3 neurons, as shown in Figure 5E and 5G. Only the CTF at 16 kDa is decreased, indicating that it is indeed a processing defect.

Additionally, the analysis of APP-CTF levels following gamma-secretase inhibition needs reassessment, as the DAPT concentration used (25µM) is excessive.

Comment 3:

3. The authors' use of 25 µM DAPT for 24 hours before analysis is problematic, as this concentration far exceeds what is necessary and may affect other aspartyl proteases (including BACE1). Given that DAPT's IC50 is in the low nanomolar range, these experiments should be repeated using 2.5 µM DAPT, which is sufficient for complete gamma-secretase inhibition. Furthermore, it's important to note that gamma-secretase inhibition can promote alpha-secretase-mediated cleavage of APP-C99 to APP-C83.

We thank the reviewer for their suggestion. Here, we chose to use 25 µM DAPT to achieve full inhibition of the gamma-secretase. This concentration and duration of treatment is in line with many papers in the literature that use a DAPT concentration of 10-50 µM, including in the following papers:

Cox *et al.*, 2025, PMID: 25316600

Feng *et al.*, 2016. PMID: 28030808

We have also shown in a previous publication that this concentration of DAPT did not affect either alpha- or beta-secretase cleavage of APP (see Januário *et al.* 2022, PMID: 35753347). In this paper, we used a RUSH-based flow cytometry assay to show that, upon treatment with 25 µM DAPT for 24 hours, there is no change in the fluorescence of an N-terminal HaloTag fused to APP, that is released upon cleavage by either alpha- or beta-secretase; however, release of a C-terminal mNeonGreen tag by gamma-secretase cleavage is completely inhibited in the presence of 25 µM DAPT. Because of this, we do not think it is necessary to repeat this set of experiments with a lower concentration, given the considerable amount of time and resources this would require. We concede however that the concentration is high, and to ensure proper interpretation by the community we have added text in the discussion highlighting this.

Interestingly, expression of wild-type RABGAP1, but not a GAP-deficient mutant, restored APP CTF (C99) levels in RABGAP1 knockdown neurons to wild-type levels. If these findings are validated by the suggested complementary analyses (comment 2), they will provide compelling evidence that RABGAP1 regulates APP trafficking and may add key information about the alterations in the cleavage mechanism.

Comment 2:

2. The current analyses would benefit from complementary measurements: i) soluble APP ectodomain fragments, and ii) total secreted amyloid-beta peptides should be considered. These data would enable a more comprehensive understanding of RABGAP1's role in APP processing.

We agree that analysis of secreted APP peptides, including both the soluble ectodomain fragment and the amyloid-beta peptides, would enhance our understanding of the role of RABGAP1 in APP processing. In response, we dedicated significant effort to optimising an ELISA assay using the commercially available Human AB42 Ultra-Sensitive ELISA Kit

(KHB3544). Despite extensive testing in both HeLa cells overexpressing dual-tagged APP and in neurons expressing endogenous APP, we were unable to detect amyloid-beta peptides with appropriate sensitivity to signal:noise. This limitation is likely due to the peptide's low abundance and rapid turnover, resulting in levels below the detection threshold of the assay. Given these technical challenges, alternative approaches such as immunoprecipitation or mass spectrometry-based quantification would likely be required to assess amyloid-beta secretion, but this is outside the scope of the present study.

In the discussion, the authors propose that RABGAP1's GAP activity inactivates RAB11A, helping maintain APP within early endosomes where proteolytic processing occurs. They hypothesize that RABGAP1 deficiency would increase membrane-associated RAB11A, causing APP accumulation in recycling endosomes where processing is less efficient. While this model provides a plausible explanation for their observations, additional experimental validation would strengthen these conclusions.

We completely agree with the reviewer that, whilst our data supports a model in which RABGAP1 regulates APP trafficking via modulation of RAB11A activity, additional experimental validation is necessary to confirm this molecular mechanism. In response, we have revised this section of the discussion to better reflect this point of view and to clearly state that future studies will be required to further interrogate the molecular details of this proposed pathway. We appreciate the reviewer highlighting this issue as we feel the new version more clearly and fairly describes the observed data.

4. The ITC data indicate that the wild-type APP tail binds to RABGAP1's PTB domain with a KD of approximately 61 μ M and a 1:1 stoichiometry. The authors should address in their discussion whether this relatively weak affinity (micromolar range) is physiologically relevant for cellular interactions.

We have now included the following text in the discussion to address this:

Using ITC, we demonstrated that the wild-type APP tail binds to RABGAP1's PTB domain with a KD of approximately 61 μ M and a stoichiometry of 1:1 (Fig. 3D). This interaction affinity is consistent with the binding affinities observed for GAP–RAB interactions, that often fall within the micromolar range, as well as for various adaptor proteins that bind APP. For example, the μ subunit of AP-4 binds to the APP tail with a KD of approximately $27.7 \pm 2.5 \mu$ M (Ross et al., 2014). These relatively weak affinity interactions are functionally important, as they allow for the dynamic and reversible trafficking of APP.

Minor comments:

A. Are HEK cells constantly treated with MycoZap, and why?

Yes, both HEK and HeLa cells are grown in MycoZap Prophylactic. This is an antibiotic used to prevent mycoplasma contamination. In addition to this, cell lines regularly undergo mycoplasma testing. There are two MycoZap products, one for decontaminating mycoplasma positive cells (MycoZap™ Mycoplasma Elimination Reagent) and one that acts as a replacement for antibiotics sometimes used in cell culture (MycoZap™ Prophylactic).

B. What is the parental cell line for the i3 neurons?

These are WTC11 human iPSCs with a stably integrated doxycycline-inducible NGN-2 transgene (Passage 22) (referred to as i3 neurons). This information has been added to the methods section of the paper. We apologise for not including this information in the submitted version- an oversight on our part.

C. Figure 4: L-M has an n = 3. Considering the variability, the authors may want to increase the number of observations to be sure about the significance.

We appreciate the reviewer's suggestion. Whilst we agree that there may be a modest increase in co-localisation in the RABGAP1 KO condition in Fig. 4L–M, we believe that even if this difference were to reach statistical significance with a larger sample size, the magnitude of the effect is likely too small to be biologically meaningful and we are hesitant to overinterpret this data. As such, we have opted not to pursue additional replicates for this analysis, but we acknowledge this limitation in our interpretation.

D. Figure 5: Please add the colocalization analysis for panel A.

We apologise that this quantification was missing from our original submission and have now included this Pearson's correlation as Figure 5B.

Reviewer 2:

Firstly, we thank the reviewer for their kind feedback and for finding our work "high quality", with "well-designed experiments that effectively support the conclusions".

1- In Figure 2: why does the FFE689-Mutant bind stronger to RABGAP1 and lose its interaction with SNX17? This is not commented on by the authors. Other mutants, such as the Y682 mutant, abolishes binding to both, RABGAP1 and SNX17. It might be good to comment on this in the main text to help the reader better grasp the story.

We are also interested in this, and we appreciate the opportunity to discuss it. The data in Figure 2 indicates that the FFE689-91 residues in the APP tail are essential for SNX17 binding, as when mutated to AAA this binding is completely abolished. Despite this, the FFE689-91AAA mutant binds to RABGAP1 with approximately a 3-fold increase. Our interpretation is that RABGAP1 and SNX17 bind competitively to the APP tail and therefore, in absence of SNX17, more RABGAP1 can bind to the APP tail in the FFE689-91AAA mutant condition compared to WT. We have now included this in the results section of the text which we hope provides clarity on this point.

2- Which Rab GTPase(s) act downstream of RABGAP1 to mediate its effects on APP trafficking? Given that the GAP-deficient RABGAP1 mutant fails to rescue the knockdown phenotype, it is likely that one or more Rab GTPases are involved. Any effort to identify or clarify this would greatly strengthen the mechanistic understanding provided by the study.

Based on the imaging data in Figure 4, along with other supporting studies, it seems likely that RAB11A is acting downstream of RABGAP1 to mediate its effect on APP trafficking. Further investigation would be required to rule out the role of other RABs in this process, however. There have been multiple previous studies that have attempted to understand RABGAP1s molecular downstream GTPases, and we have included this in the discussion.

3- What is the identity of the compartment where RABGAP1 colocalizes with APP? Including experiments with specific organelle markers and quantifying the co-localization would improve the clarity and robustness of this finding.

In response to the reviewer's suggestion, we attempted to quantify the colocalisation of APP and RABGAP1. However, this proved to be technically challenging due to the predominantly cytosolic localisation of RABGAP1, as acknowledged in the main text in the original submission and consistent with previous descriptions. Given this limitation, we instead examined the colocalisation of APP with RAB11A and RAB5A, which are established markers for recycling endosomes and early endosomes, respectively. Our findings demonstrate a clear colocalisation of APP with these endosomal markers, consistent with previous findings which support its localisation to the endosomal system under steady-state conditions. These results are consistent with numerous previous studies that have reported similar APP localisation patterns.

4- How does the localization of the double-tag APP in neurons compare to the localization of endogenous APP?

The use of a dual-tagged APP fusion protein in HeLa cells has already been characterised in Januário et al. (2022). Here, it is shown that the steady state distribution of APP is comparable to WT, or any differences fall below the detection limit, in the presence of the two tags and that dual-tagged APP undergoes physiological processing by the secretase enzymes.

In Figure 5A, APP localisation to the early endosomes in iNeurons is shown using overexpression of both HaloTag-APP-mNeonGreen and mScarlet-FYVE. In line with the HeLa cell data, APP and FYVE colocalise, indicating that APP is present in the endosomes in neurons. This is in line with many previous papers that demonstrate endogenous APP largely localises to the early endosomal system.

5- Figure 5F&G: I might have misunderstood this, but why does RABGAP1 knockdown reduce the levels of C99, while at the same time not affecting the amounts of full-length APP. In Figure 4, the KO of RABGAP1 increased full length APP as a sign of reduced processing. What explanations do the authors have for these findings.

The reviewer is correct in pointing out that in iNeurons, the levels of C99 is significantly reduced, whereas the levels of full length APP appeared to be unaffected. Since the decrease in C99 abundance is still observed in the RABGAP1 KO cells when treated with the gamma secretase inhibitor, DAPT, this suggests that the defect is in beta-secretase cleavage of APP in the KO cells, rather than subsequent gamma secretase cleavage.

This is indeed different to the phenotype observed in the RABGAP1 KO HeLa cells, where levels of both N-terminal HaloTag and C-terminal mNeonGreen are significantly increased, perhaps suggesting that full length APP levels are increased. We do not detect a significant increase in full length APP levels in the iNeuron system, however. Whilst the phenotypes appear slightly different, both suggest a defect in the initial cleavage step by the beta-secretase.

One explanation for this discrepancy may be that APP processing in the two systems is significantly different. In HeLa cells, the expression levels of the secretase enzymes are very different in HeLa cells compared to neurons, and in one system we look at overexpressed APP whereas in the other we are monitoring endogenous APP processing. We think this highlights the importance of using both multiple and more relevant cell types. We also wish to highlight that this discrepancy in processing pathways in different cell models is consistent with expectations and does not affect our ultimate mechanistic conclusions.

Reviewer 3:

In my opinion, the main issue is whether the evidence suffices to conclude that the observed perturbation of APP trafficking and processing is indeed due to a direct interaction between the APP tail and RABGAP1, or else, whether the perturbation is indirect, i.e. caused by the perturbation of trafficking that is well known to occur upon RABGAP knockout. Deleting RABGAP1 is known to result in multiple cellular phenotypes that are supposed to be caused by impaired GTP-cleavage of Rab6 (or Rab11), known regulators of Golgi-endosome trafficking. In fact, the observation that the GTPase activating function is required for the APP effects supports the second view: As far as I can see, there is no evidence showing that the interaction between the two proteins is functionally required for the observed effects on APP processing (which is acknowledged by the authors). Thus, the manuscript largely confirms (and maybe extends) the notion that perturbation of post-Golgi trafficking (via specific Rabs) affects APP trafficking and processing, and it is unclear whether there is any specific additional function of RABGAP1 as insinuated.

We agree that it is challenging to ascribe direct effects in membrane trafficking systems. The evidence we have for direct binding, we feel, is very strong as it comes from both unbiased lysate precipitations, validation with endogenous prey, biochemical interaction mapping, *in silico* interaction mapping, and finally *in vitro* direct binding assessment. We also feel confident (as the reviewer supports) that there is an APP processing phenotype upon abrogation of RABGAP1, which we have shown in multiple cell systems and cell models. The key piece of evidence, we wish to highlight to the reviewer, that the binding is affecting the phenotype is Figure 6A-B where we mutate the validated RABGAP1-APP binding site on RABGAP1 and show that complementation does not recover the phenotype. We consider this experiment a key piece of data that goes beyond most molecular cell biology studies with similar claims and represents a gold standard.

However, as the reviewer states and as we stated in the original manuscript, there could be other interpretations. For example, that same binding site on RABGAP1 could be for multiple cargoes. To address these issues the reviewer rightly highlights, we have made two changes.

Firstly, we have clearly highlighted this further in the discussion.

Secondly, we have set out to demonstrate that there is not a major broad disruption to the membrane trafficking in cells leading to vast off-target effects. To do this, we have performed a set of experiments to assess the constitutive secretory pathway, which is one of the core expertise of our lab. We have previously demonstrated using a highly quantitative RUSH-based secretory pathway assay (Pereria 2023) that cell surface delivery of cargo is highly sensitive to RAB6 and thus can, in this instance, act as a readout for general trafficking. In brief, we have transiently knocked out RABGAP1 and, for extra assurance, its homolog RABGAP1L, or both, in the context of our trafficking assay. In support of our findings, there is a very minor defect in the secretory pathway. This data has been added to the manuscript as Figure S10.

We thank the reviewer for giving us the opportunity to focus on this as it is a classic, difficult and important issue. We feel that these additions have strengthened our findings and allowed us to clarify the strength of evidence.

a) Fig. 4A-C: The authors compare fluorescence intensities between different cell lines (wt, KO) but there is no standardization, e.g. IC for an endogenous protein. In Fig. B it appears that for some reason more cells were used for the KO than for the wt control (the tubulin band is quite a bit stronger), showing that calibration is needed to unequivocally document the alleged increase in expression level.

We apologise for the confusion here, which we take responsibility for, and we thank the reviewer for the opportunity to clarify it for the readership. In short: Figure 4B and 4C/D are not related experiments/quantifications. In Figure 4B, there are indeed higher tubulin levels which is due to loading and works in the favour of the claim that the KO is successful. Fig. 4C-D are flow cytometry experiments where the KO protein levels are normalised to the WT as a geometric mean of the population. We apologise for this issue and have addressed it by 1) changing the axis label, 2) changing the figure legend to ensure it is clear to a reader.

Generally, the characterization of the RABGAP1 and APP positive structures is insufficient - what are these dots? Did the authors quantify the degree of colocalization?

The degree of colocalisation between APP and the endosomal markers RAB11A and RAB5A was quantified in Figure 4, supporting the interpretation that the APP-positive puncta represent distinct classes of endosomes. These findings are consistent with previous literature characterising APP trafficking through the endosomal system. As noted in the manuscript, and discussed in the response to the reviewer above, quantifying colocalisation between APP and RABGAP1 was particularly challenging due to the predominantly cytosolic localisation of RABGAP1 and its transient recruitment to endosomal membranes. This dynamic behaviour limits reliable spatial overlap with APP

in fixed samples. Furthermore, the endosomal localisation of RABGAP1 has been more extensively characterised in a previous study (Samarelli et al., 2020), and we have therefore focused our current work on the functional consequences of RABGAP1 loss, rather than repeating their well-performed and detailed localisation analyses.

b) Fig. 4 H-M. Here the authors quantify colocalization of APP with Rab11 and Rab5 respectively. I have several problems with these experiments. First, all data are based on overexpressed and tagged proteins and thus do not represent the native state. Second, even when pushing the expression levels the degree of colocalization appears to be small (cross-correlation coefficients between 0.1 and 0.15), questioning the significance of the overall rather small changes.

We agree that the use of overexpressed, dual-tagged APP in HeLa cells does not represent its native physiological context. This system was chosen for our initial experiments because it allows for robust, quantitative detection of changes to APP cleavage under different conditions. While this approach is widely used in the field to monitor APP dynamics, we fully acknowledge its limitations. For this reason, we subsequently transitioned to a neuronal cell model to study endogenous APP processing under more physiologically relevant conditions, as shown in the western blotting experiments in both iNeurons and primary rat hippocampal neurons. This allowed us to validate and extend our findings beyond the overexpression HeLa cell system. To ensure that readers are aware of these limitations, we have added this caveat to the discussion.

In addition, there is no information about cell-to-cell variability, about transfection efficiency, about expression levels in the transient expression experiments that I could find in the methods section. How was the statistics done? It makes no sense to calculate significances with n=3 (!). Also, were these independent experiments, or just different cells from the same transfection that were analysed?

We thank the reviewer for raising this point. The Figure 4 legend has now been expanded to more clearly state that “N=3” refers to the number of independent biological replicates. Each point on the bar graphs represents the average from one biological replicate and a minimum of 28-30 cells were analysed in total in each condition. To avoid confusion, we have now clarified this more explicitly in the figure legend. We also wish to note that we consider it proper to report independent repeats rather than technical, and we consider the statistical quality of our findings not only in line with but superseding the current standard in cell biology. The statistical analysis shown follows standard practice for image quantification in cell biology, where significance is assessed across biological replicates using subsequent statistical tests. We hope this clarification addresses the reviewer’s concern.

c) The supplementary videos lack a description (imaging conditions, frame rate, labelling etc.).

We apologise if the following supplementary video legends were not uploaded correctly with the data, which we take responsibility for, and have amended here:

Supplementary Video 1:

Co-localisation of APP and RABGAP1 in HeLa cells pre-treated with 25 μ M DAPT for 24 hours. Live cell structured illumination microscopy of steady state HaloTag-APP-mNeonGreen and mCherry-RABGAP1. Videos were acquired on a Zeiss Elyra 7 Lattice SIM microscope. Scale bar, 5 μ m. Frame rate, 7 fps (frames per second)

Supplementary Video 2:

Inset from Supplementary Video 1. Co-localisation of APP and RABGAP1 in HeLa cells pre-treated with 25 μ M DAPT for 24 hours. Live cell structured illumination microscopy of steady state HaloTag-APP-mNeonGreen and mCherry-RABGAP1. Videos were acquired on a Zeiss Elyra 7 Lattice SIM microscope. Scale bar, 5 μ m. Frame rate, 7 fps

Supplementary Video 3:

Co-localisation of APP and endosomes in day 15 WTC11 human-derived i3 neurons. Live cell imaging of steady state HaloTag-APP-mNeonGreen and mScarlet-FYVE. Images were taken on a Zeiss LSM880 Airyscan microscope. Scale bar, 10 μ m. Frame rate, 7 fps

Dear David,

We have now received re-review reports from two referees, which I have included below. As you will see, you have addressed their concerns satisfactorily. Before I can finally accept the manuscript, there are some remaining editorial points which need to be addressed. In this regard would you please:

- upload the manuscript in a .docx format, switching off track changes,
 - acknowledge all funding in our online submission system (not in the Comments box),
 - label the 'keywords' section,
 - rename the bibliography "References",
 - include a section title for the abstract,
 - rename the conflict of interest statement as the "Disclosure and competing interests statement",
 - include references to Figures 2C-E, S4, S7, S8 in the manuscript text,
- remove figures from the manuscript file, listing figure legends below the References,
- update source file names, title, legend and manuscript callout for Dataset EV1 (instead of Appendix Table S1/Supplementary Table 1); the legend should be uploaded as a separate tab/sheet in the Excel file,
 - use "Appendix for RABGAP1 acts as a sensor to facilitate sorting and processing of amyloid precursor protein" in the appendix title page and include a table of contents with the page numbers for the listed items; the nomenclature should be Appendix Figure Sx and Appendix Table Sx throughout the manuscript and Appendix PDF,
 - include a table of Resources and Tools,
 - remove "Supplementary Information" with all Appendix sections from the manuscript file and only listed in Appendix PDF,
 - rename movie files as 'Movie EV1-EV3' with the corresponding callouts, zipping the legends with each movie file,
 - define p values in the legends of figures 2C-E; 4C, D, E, I, J, 5F, H, I; 6B, D,
 - indicate the statistical test used for data analysis in the legend of figure 1D,
 - define box plots in terms of minima, maxima, centre, bounds of box and whiskers, and percentile in the legends of figures 4C, D,
 - define the nature of n in the legends of figures 4E, M; 5I,
 - define error bars in the legends of figures 2C-E; 4E, F, I, J, L, M; 5B, C, F, G, H, I; 6B, D, and
 - correct the section order as follows: Title page - Abstract & Keywords - Introduction - Results - Discussion - Methods - Data Availability - Acknowledgements - Disclosure and Competing Interests Statement - References - Figure Legends - Table(s) - Expanded View Figure Legends.

We include a synopsis of the paper (see <http://emboj.embojournal.org/>). Please provide me with a general summary image, a two sentence statement and 3-5 bullet points that capture the key findings of the paper.

I am looking forward to receiving your revised manuscript.

EMBO Press is an editorially independent publishing platform for the development of EMBO scientific publications.

Best wishes,

William

William Teale, PhD
Editor
The EMBO Journal
w.teale@embojournal.org

Referee #2:

The authors have responded well to all my previous comments. I went also through the response to the other reviewers and think that the addition of new data further strengthened the story. I have no further comments.

Referee #3:

During revision, the authors addressed all points raised in my report, thus alleviating my main concerns. Thus, I am not opposed to publication of the manuscript (also considering that the other reviewers were more positive). Just one more comment about the statistics: My aim was not to question whether the noted differences in the various plots are convincing or not - as a

biochemist/cell biologist I know well that even three repetitions involve a lot of work, and if the data are clear, it is perfectly acceptable (and I am not only referring to Fig. 4). Calculating significances based on $n=3$, however, is not sensible, even though it still is commonplace among cell biologists - it does not make it any better. I acknowledge that the individual data points are depicted in the plots. Thus the data of the individual experiments are presented, and insofar I do not object if the authors insist on using these methods.

All editorial and formatting issues were resolved by the authors.

Dear David,

I am pleased to inform you that your manuscript has been accepted for publication in the EMBO Journal.

Congratulations! I am delighted that this study will be published with us.

Best wishes,

William

William Teale, PhD
Editor
The EMBO Journal
w.teale@embojournal.org
